# Learning Augmentation Distributions using Transformed Risk Minimization

**Evangelos Chatzipantazis**[*]                *vaghat@seas.upenn.edu*
*Department of Computer and Information Science*
*University of Pennsylvania*

**Stefanos Pertigkiozolou**[*]                *pstefano@seas.upenn.edu*
*Department of Computer and Information Science*
*University of Pennsylvania*

**Kostas Daniilidis**                   *kostas@cis.upenn.edu*
*Department of Computer and Information Science*
*University of Pennsylvania*

**Edgar Dobriban**                *dobriban@wharton.upenn.edu*
*Department Statistics of Statistics and Data Science*
*University of Pennsylvania*

**Reviewed on OpenReview:** *https://openreview.net/forum?id=LRYtNj8Xw0*

## Abstract

We propose a new *Transformed Risk Minimization* (TRM) framework as an extension of classical risk minimization. In TRM, we optimize not only over predictive models, but also over data transformations; specifically over distributions thereof. As a key application, we focus on learning augmentations; for instance appropriate rotations of images, to improve classification performance with a given class of predictors. Our TRM method (1) jointly learns transformations and models in a *single training loop*, (2) works with any training algorithm applicable to standard risk minimization, and (3) handles any transforms, such as discrete and continuous classes of augmentations. To avoid overfitting when implementing empirical transformed risk minimization, we propose a novel regularizer based on PAC-Bayes theory. For learning augmentations of images, we propose a new parametrization of the space of augmentations via a stochastic composition of blocks of geometric transforms. This leads to the new *Stochastic Compositional Augmentation Learning* (SCALE) algorithm. The performance of TRM with SCALE compares favorably to prior methods on CIFAR10/100. Additionally, we show empirically that SCALE can correctly learn certain symmetries in the data distribution (recovering rotations on rotated MNIST) and can also improve calibration of the learned model.

## 1 Introduction

Adapting to the structure of data distributions (such as symmetry and transformation invariances) is an important problem in machine learning. Invariances can be built into the learning process by architecture design (such as in convolutional neural nets), or by augmenting—transforming—the dataset during training. Traditionally, these approaches require full knowledge of the symmetries of the data distribution. Absent this knowledge, practitioners may mis-specify the set of augmentations, or may need to resort to expensive and time-consuming tuning. The current practice of machine learning largely depends on handcrafted augmentations, based on extrinsic knowledge about the task to be solved. In contrast, the "best" augmentations

---

[*]Equal Contribution.

may also depend on the predictive model, not just on the task; in possibly non-obvious ways. The space of augmentations is vast; and these factors make human fine-tuning hard. Therefore, it could be more efficient to learn augmentations from data. There have been several prior approaches to this problem, but none that satisfy our desired criteria described below.

Our goal is to design an algorithm that learns data transformations jointly with solving a downstream task. For instance, we want to learn which augmentations—rotations or flips—are most helpful while learning to classify images. In addition, we want to learn the level of data transformations—maximum rotation angle—that will be beneficial for the task. We require that the algorithm is:

1. *Efficient*: The goal is to jointly learn augmentations and models in a *single training loop* (without a significant increase in training time compared to standard model training).
2. *Modular*: The method should pair with any learning problem and algorithm for the downstream task. For example, it should be independent of the neural network architecture in an image classification task and should be incorporated without significant changes into a standard training algorithm for risk minimization.
3. Applicable to any transforms, for instance to both continuous and discrete-valued augmentations. This is challenging, as previous approaches have used specialized approaches (Cubuk et al., 2019a; Benton et al., 2020): for discrete augmentations, one can sometimes enumerate them; for continuous ones one may leverage Lie group theory.

Our approach to this problem (in Section 2) is, intuitively, to expand the usual model space (e.g., neural networks) to include a first "stochastic layer" where we optimize over *distributions of transformations* using a novel objective. In the framework of statistical learning theory, the transformations induce a so-called *transformed risk*. Thus, we formulate our algorithm in a new theoretical framework called *Transformed Risk Minimization* (TRM). This enables us to leverage the theory of risk minimization and design a unified algorithm that satisfies all three criteria discussed above. We can summarize our contributions as follows:

1. We formulate the Transformed Risk Minimization framework (TRM) and connect it to standard risk minimization. We prove a zero-gap theorem, implying that when there is an unknown *distributional invariance* in the data, TRM can recover both the invariance as well as a model that is optimal under the standard risk when the distributional invariance is known.
2. We then propose the new Stochastic Compositional Augmentation Learning (SCALE) algorithm that aims to optimize the transformed risk for finite data. SCALE consists of a new parametrization of the augmentation space via a stochastic composition of blocks and a novel regularizer for training which is derived using PAC-Bayes theory.
3. We provide experiments with SCALE. Our first experiment shows that on RotMNIST, SCALE learns the correct symmetries and outperforms previous methods; which supports our zero-gap theorem. We also perform experiments on CIFAR 10 and CIFAR 100, showing that SCALE has advantages in both accuracy and calibration compared to prior works, while maintaining the benefit of time efficiency.

## 2 Transformed Risk Minimization

### 2.1 Preliminaries on Risk Minimization

In conventional supervised statistical learning (Vapnik, 2013), the learner is provided with an independently drawn sample $S = \{(X_i, Y_i), i \in [n]\}$, where $[n] = \{1, 2, \ldots, n\}$, and $(X_i, Y_i) \in \mathcal{X} \times \mathcal{Y}$ for all $i \in [n]$, from an unknown data distribution $\mathcal{D}$. Further, the learner has access to an action space $\mathcal{Y}'$,[1] and a hypothesis space $\mathcal{H} \subset (\mathcal{Y}')^{\mathcal{X}}$ containing candidate hypotheses (models) $h : \mathcal{X} \to \mathcal{Y}'$. In general, $\mathcal{Y}'$ might not be equal to $\mathcal{Y}$. For example, in a multi-class classification problem, $\mathcal{Y}$ can be the discrete set of classes, and $\mathcal{Y}'$ can be the set of probability distributions over them.

---

[1]All objects are measurable with respect to appropriate sigma-algebras, which are used tacitly in our work.

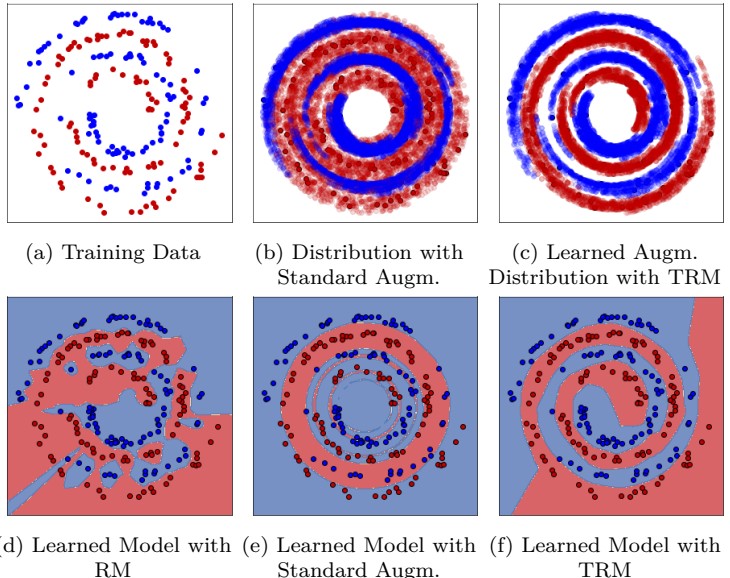

(a) Training Data
(b) Distribution with Standard Augm.
(c) Learned Augm. Distribution with TRM

(d) Learned Model with RM
(e) Learned Model with Standard Augm.
(f) Learned Model with TRM

Figure 1: When only a small amount of training data is available (as shown in (a)), models trained with standard risk minimization (RM) may lead to poor generalization. A common practice to improve the generalization is to perform data augmentation, but this can be harmful when the appropriate distribution of augmentations is not known (as shown in (b), (e) where we apply rotations of 180°). We propose the framework of Transform Risk Minimization (TRM) to learn both the model and the optimal distribution of augmentations. As shown in (c), (f), TRM recovers the appropriate distribution of augmentations (rotations of 45°) and learns an improved model.

Based on the training data $S$, the learner outputs a hypothesis $h_S$ whose performance on a test datapoint $(x, y) \sim \mathcal{D}$ is evaluated via a loss function $l : \mathcal{Y}' \times \mathcal{Y} \to \mathbb{R}$. The goal in Risk Minimization (RM) is to minimize the following risk over $h \in \mathcal{H}$:

$$R_{RM}(h) := \mathbb{E}_{(X,Y)\sim\mathcal{D}}[l(h(X), Y)]. \tag{1}$$

One may aim to control the risk $R_{RM}(h_S)$ of the data-dependent predictor $h_S$ either in expectation or with high probability with respect to $S$.

## 2.2 From fixed augmentations to an augmentation hypothesis space

A common way to improve performance is by leveraging domain knowledge in the form of transformations that leave the prediction invariant (e.g., in classification), or equivariant—transforming in a pre-specified way (e.g., in pose estimation of complex shapes). When the set of transformations admits a group structure, it is beneficial to constrain the hypothesis space to invariant/equivariant models. However, for more general transformations, the most common practice is *data augmentation*: adding transformed data to training. If tuned properly, data augmentation can improve performance. However, it assumes that the user fully knows the beneficial transformations. This assumption is in general not valid; and in practice augmentations require a great deal of tuning.

To relax this assumption, we switch our viewpoint and suppose that the user only has a certain level of *incomplete knowledge* about the beneficial transformations. We then aim to learn those transformations *jointly* with the model. Intuitively, one may think that one should optimize over specific transformations $g : x \mapsto g(x)$. However, we find it beneficial to relax to *distributions* over transformations, as this is both more general—it includes point masses at specific transformations—and more closely mirrors standard stochastic data augmentation. Thus, we consider a modification of the classical learning paradigm in which an additional *transformation hypothesis space* $\Omega$ is provided to the learning algorithm, containing candidate *transformation distributions* $Q$ over sets of transformations $G_Q$. Then, we denote the action of $g \in G_Q$ on

$X$ as $gX$ (instead of $g(X))^2$. Now the goal of the learner is to output not just a model $h_S \in \mathcal{H}$, but also a distribution $Q_S \in \Omega$ of transformations.

As an example, one transformation distribution $Q \in \Omega$ might be the uniform distribution over all rotations in the range $(-30°, 30°)$ degrees, i.e., $Q = \mathrm{U}_R(-30°, 30°)$.[3] The distribution hypothesis space $\Omega$ could be the set of all such uniform distributions parametrized by their endpoints, e.g., $\Omega = \{\mathrm{U}_R(\theta_1, \theta_2) \,|\, \theta_1 < \theta_2, |, \theta_1, \theta_2 \in [-180°, 180°]\}$. The sampled transforms $g \sim \mathrm{U}_R(-30°, 30°)$ act on their inputs by rotation. For 2D image inputs, they are represented by specific $2 \times 2$ rotation matrices transforming an image $I : \mathbb{R}^2 \to \mathbb{R}$ as $I(x_1, x_2) \xrightarrow{g} gI(x_1, x_2) := I(g^{-1}(x_1, x_2))$. In this example, we learn the range of the helpful rotations, as well as a model that fits the transformed data.

## 2.3 Transformed Risk Minimization

A first thought may be to try to optimize the standard risk in equation 1 for the *expected* prediction over the unknown $Q \in \Omega$, i.e., $\bar{R}(h, Q) := \mathbb{E}_{(X,Y)\sim\mathcal{D}}[l(\mathbb{E}_{g\sim Q}[h(gX)], Y)]$. However, this objective is not amenable to standard stochastic optimization due to the difficulty of obtaining unbiased gradients with respect to $Q$.

In this work we study a notion of *transformed risk* that bypasses this issue. The goal of the learner in Transformed Risk Minimization (TRM) is to minimize the *transformed population risk*

$$R_{TRM}(h, Q) := \mathbb{E}_{(X,Y)\sim\mathcal{D}}\mathbb{E}_{g\sim Q}[l(h(gX), Y)] \tag{2}$$

simultaneously over augmentation distributions $Q \in \Omega$ and models $h \in \mathcal{H}$. Figure 1 shows an example of the difference between models learned using the standard risk minimization and our proposed TRM framework.

This objective is inspired by how data augmentation is applied in practice with a fixed distribution $Q$ (Baird, 1992; Chen et al., 2020). In particular, data augmentation can be viewed as averaging the loss over the augmentation distribution. In our setting, this distribution is learned. For losses such that $l(\cdot, y)$ is convex for all $y \in \mathcal{Y}$, such as typical surrogate losses for classification like the cross-entropy loss, Jensen's inequality shows that $\bar{R}(h, Q) \leq R_{TRM}(h, Q)$. Thus a small value of the TRM objective implies that the more difficult-to-handle risk of the average model is also small. Given a finite dataset, empirical transformed risk minimization (ETRM) would aim to optimize:

$$R_{TRM,n}(h, Q) = \frac{1}{n} \sum_{i=1}^{n} \mathbb{E}_{g\sim Q} l(h(gX_i), Y_i), \tag{3}$$

over $h \in \mathcal{H}, Q \in \Omega$. However, as we discuss in Section 3.2 this training objective is not a good estimate of the population risk (equation 2) as it collapses to trivial distributions. In the next section, we discuss how to optimize the population risk by proposing an upper bound to equation 2 that is computable from the training data and takes care of this problem.

# 3 The SCALE Algorithm for Learning Data Augmentations

In this section, building on our TRM framework, we design the new *Stochastic Compositional Augmentation Learning* (SCALE) algorithm, shown in Figure 2.

## 3.1 Parametrization of $Q$ and $\mathcal{H}$

In practice, we do not optimize over an abstract space $(\mathcal{H}, \Omega)$ but over a parametric space $(W, \Theta)$. Considering the complexity of our tasks, we will use rich hypothesis spaces $\mathcal{H}$—such as deep networks—with parameters $w$ for some parameter space $W \subseteq \mathbb{R}^p, p \in \mathbb{N}$, i.e., $\mathcal{H} = \{h_w; w \in W\}$.

To parametrize the augmentation distributions $Q$ in $\Omega$, we leverage the rich compositional structure of transformations. Complex transformations can be constructed by compositions of simpler transformations sampled from parametrized base distributions $\{U_{i,\alpha_i}\}_{i=1}^{k}$. We can also include in the composition fixed base

---

[2]In general, $g : \mathcal{X} \to \mathcal{X}', h : \mathcal{X}' \to \mathcal{Y}'$. We simplify $\mathcal{X}' = \mathcal{X}$ for clarity.

[3]The symbol U will denote a uniform distribution over an appropriate space.

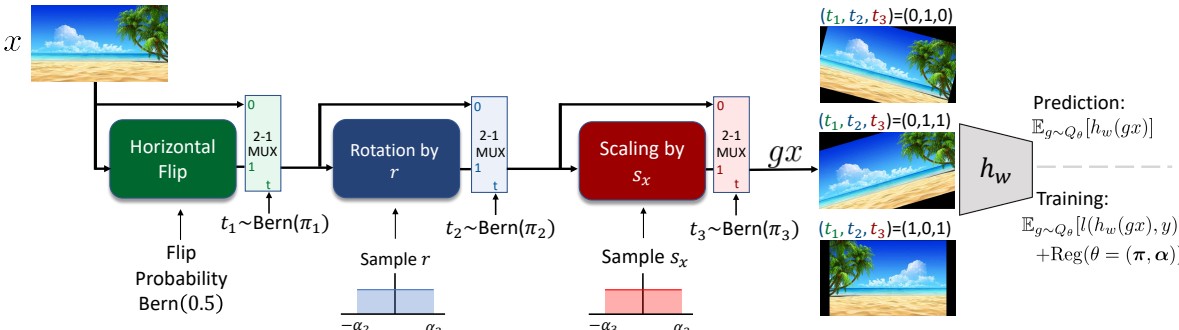

Figure 2: SCALE: The computation of $gx$ from a random sample $g \sim Q_\theta$ is a composition of blocks applying randomly sampled transformations to the input. The $i^{th}$ block in the sequence transforms its input with probability $\pi_i$ and leaves it unchanged with probability $(1 - \pi_i)$, described in the figure via 2-1 multiplexers "MUX" with probabilistic selectors. For each input $x$, the model samples several $gx$-s and uses the outputs $h(gx)$ to compute the expected loss during training, and the expected output at test-time inference. During training, we learn the parameters of both the model and the distribution of augmentations by minimizing the expected loss plus the regularization term we derive in Section 3.2.

distributions $\{U_i\}_{i=k+1}^{K}$. Additionally, we allow choosing which base distributions should be included in the composition independently for each $i$. This is a discrete optimization problem where the optimization variable can take an exponential number of values in the number of base distributions. We associate to each base distribution a variable $\pi_i \in \{0, 1\}$ indicating if the distribution is included in the composition. To handle the combinatorial optimization problem, we relax $\pi_i$ to continuous mixture components $\pi_i \in [0, 1]$, $i \in [K] := \{1, \dots, K\}$. Thus, we define $Q_\theta$ to be the distribution induced by sampling $g \sim Q_\theta$ as:

$$g = g_1 \circ g_2 \circ \dots \circ g_K, \quad g_i \sim Q_{(i)}, \quad i = 1, \dots, K \tag{4}$$

where $\theta = \{\pi_1, \dots, \pi_K, \alpha_1, \dots, \alpha_k\}$ and $g_i \sim Q_{(i)}$ is:

1. with probability $\pi_i$ a transformation sampled from a distribution $U_{i,\alpha_i}$. This distribution can have learnable parameters $\alpha_i$, or can be fixed (the latter denoted as $U_i$);
2. with probability $(1 - \pi_i)$ the identity transformation leaving the input unchanged.

With this parametrization we can handle compositions of both discrete and continuous distributions of transformations. This is an advantage compared to previous works, as we discuss in the Related Work section.

We will focus on base augmentation distributions $U_{i,\alpha_i}$ supported on transformations of the form $\rho_i(a)$, with $a$ sampled from the uniform distribution parametrized by $\alpha_i$ via $a \sim \mathrm{U}([-\alpha_i, \alpha_i])$, where $0 < \alpha_i \le A_i$ for some fixed $A_i > 0$.[4] We choose a parametrization of our transforms such that $\rho_i(0)$ is the identity map. For instance, $\rho_i(a)$ can be a rotation by $a$ degrees with $a$ sampled uniformly over the interval $[-30°, 30°]$. We will also consider parameter-free base distributions $U_i$, uniform on some discrete set containing the identity transformation (such as horizontal flips, or cropping). If the first $k$ out of $K$ base distributions are parametric, then the base mixed distributions $Q_{(i)}$[5] are constructed as

$$Q_{(i)} = \begin{cases} (1 - \pi_i)\delta_0 + \pi_i \mathrm{U}[-\alpha_i, \alpha_i], & i \le k \\ (1 - \pi_i)\delta_0 + \pi_i \mathrm{U}[\{0, \dots, N_i - 1\}], & k < i \le K, \end{cases}$$

where $\alpha_i \in \Theta_i := (0, A_i]$ and $\pi_i \in [0, 1]$ are the parameters of the mixture we optimize over. Then, $\Theta = [0, 1]^K \times \prod_{i=1}^{k} \Theta_i$, and for each $\theta \in \Theta$, each $Q_\theta$ is constructed as the product measure $Q_\theta = \prod_{i=1}^{K} Q_{(i)}$

---

[4]Non-symmetric endpoints are handled similarly.

[5]We overload notation here. $Q_{(i)}$ was supported on a set of transformations $\rho_i(a)$ not the parameters $a$. We explain this common subtlety and how it can lead to tighter bounds in the Appendix.

on $\prod_{i=1}^{k}[-a_i, a_i] \times \prod_{i=k+1}^{K}\{0, \ldots, N_i - 1\}$. Finally the augmentation hypothesis space is $\Omega = \{Q_\theta : \theta \in \Theta\}$. Distributions of transformations where the parameter is sampled over a closed interval $[-\alpha_i, \alpha_i]$ are common in practical data augmentation. However, in contrast to their standard usage, here we can learn the endpoints.

The task is to learn parameters $(w, \theta) \in (W, \Theta)$ such that $(h_w, Q_\theta) \in (\mathcal{H}, \Omega)$ results in a small TRM risk 2.

## 3.2 Leveraging PAC-Bayes theory for training SCALE

In this section we design an algorithm aiming to optimize the transformed risk given a finite sample. Naively optimizing the empirical TRM criterion (Eq. 3) would typically result in a trivial solution for the augmentation distribution $Q_\theta$, namely the Dirac mass around the identity transformation. This is not surprising, since augmenting the initial dataset typically results in worse training error, even though it might result in better generalization; see e.g., Benton et al. (2020) etc.

Thus, regularization is of high importance for identifying the useful augmentations. We propose to leverage PAC-Bayes theory (McAllester, 1998; Shawe-Taylor & Williamson, 1997; Catoni, 2007) to bound the TRM risk and to obtain a novel, theoretically motivated regularizer.

Due to space limitations we will assume some basic familiarity with PAC-Bayes theory. We provide a bound on SCALE building on a PAC-Bayes bound (Alquier et al., 2016) that holds for bounded and some unbounded losses (e.g., sub-gaussian losses (Germain et al., 2016b)). Since we aim for generality and efficiency in our algorithm, we do not use bounds from (McAllester, 2013)—derived only for bounded losses—or data-dependent priors (Dziugaite & Roy, 2018)—which usually require nested training loops. In our experiments, our regularizer is essential for recovering useful augmentation distributions; even though the bounds themselves may not provide a non-vacuous generalization certificate. After all, we use deep networks for which non-vacuous bounds are uncommon.

Our generalization bound relies on the following reduction: instead of viewing data augmentation as a transformation of the data followed by the application of a hypothesis $h \in \mathcal{H}$, we view the entire process as a randomized predictive model. For each $(h, Q)$, the transformation distribution $Q$ acts as a first "stochastic layer" before applying a deterministic model $h$. Hence, Eq. 2 becomes the expected risk of a stochastic classifier $\tilde{h} = h \circ g$, $g \sim Q$. For each input $x$, the classifier $\tilde{h}$ first draws a sample $g \sim Q$ and then predicts $(h \circ g)(x)$.

Let $P_{\mathcal{H}} = \mathcal{N}(w_0, s^2 I)$, $Q_{\mathcal{H},w} = \mathcal{N}(w, \sigma^2 I)$, for some parameters $w_0, w \in W$ be multivariate normal distributions with scaled identity covariance matrices. Let $Q_\theta$ be the probability measure constructed in Section 3.1 and, similarly to $Q_\theta$, let $P$ be the product measure $P_A = \prod_{i=1}^{k} P_{(i)}$ on $A := \prod_{i=1}^{k}[-A_i, A_i] \times \prod_{i=k+1}^{K}\{0, \ldots, N_i - 1\}$, where

$$P_{(i)} = \begin{cases} \beta\delta_0 + (1-\beta)\mathrm{U}[-A_i, A_i], & i \leq k \\ \beta\delta_0 + (1-\beta)\mathrm{U}[\{0, \ldots, N_i - 1\}], & k < i \leq K \end{cases}$$

for a fixed $\beta \in (0, 1)$. In practice, we choose $\beta$ to be a small positive number.[6] Consider the prior and posterior probability measures $P_{\mathcal{H}} \times P_A$ and $Q_{\mathcal{H},w} \times Q_\theta$ on $W \times A$. Then, the following bound holds for the TRM risk:

**Theorem 3.1** (SCALE generalization bound). *If the map $w \to l(h_w(X), Y)$ is $L$-Lipschitz continuous for any $(X, Y) \in \mathcal{X} \times \mathcal{Y}$ and $l$ is bounded in $[a, b]$,[7] then, with probability at least $1 - \delta$ over the draw of $S = \{(X_i, Y_i)\}_{i=1}^{n} \sim \mathcal{D}^n$ it holds simultaneously over all $w \in W$, $\theta = \{\alpha_1, \ldots, \alpha_k, \pi_1, \ldots, \pi_K\} \in \Theta$ where $\alpha_i \in (0, A_i], i \in [k]$ and $\pi_i \in [0, 1], i \in [K]$ that:*

$$R_{TRM}(h_w, Q_\theta) \leq R_{TRM,n}(h_w, Q_\theta) + \frac{1}{\sqrt{n}}\left(\frac{1}{2s^2}\|w - w_0\|^2 + \mathrm{Reg}(\theta)\right) + c_n$$

---

[6]$\beta > 0$ is necessary so that $Q \in \Omega$ is *absolutely continuous* with respect to $P_A$ and $KL(Q\|P_A)$ is well-defined.

[7]In the Appendix we provide a more general bound for any loss satisfying a so-called *Hoeffding assumption* (Alquier et al., 2016).

*where*

$$\text{Reg}(\theta) := \sum_{i=1}^{k} K_i(\alpha_i, \pi_i) + \sum_{i=k+1}^{K} K_i(\pi_i). \tag{5}$$

*Here, for $i \leq k$,*

$$K_i(\alpha_i, \pi_i) = \log \frac{1 - \pi_i}{\beta} + \pi_i \log \left[ \frac{\pi_i A_i \beta}{(1 - \pi_i)\alpha_i(1 - \beta)} \right] =$$

$$\text{KL}\left(\mathcal{B}(\pi_i) || \mathcal{B}(1 - \beta)\right) + \pi_i \text{KL}\left(\text{U}[-\alpha_i, \alpha_i] || \text{U}[-A_i, A_i]\right)$$

*and for $k < i \leq K$,*

$$K_i(\pi_i) = \text{KL}\left( \mathcal{B}\left( \left(1 - \frac{1}{N_i}\right) \pi_i \right) \,\middle\|\, \mathcal{B}\left( \left(1 - \frac{1}{N_i}\right)(1 - \beta) \right) \right).$$

*and $c_n = c_n(\delta, s, p, L) = o_n(1)$ vanishes as $n \to \infty$.*

The proof is provided in Appendix A.2. The steps are as follows:

1. **PAC-Bayes for joint bound:** Using our alternative viewpoint discussed above, we leverage PAC-Bayes theory to derive generalization bounds on the expected risk jointly over models and transformations.
2. **Derandomization:** However, in TRM we are interested in deterministic models. Thus, we apply the Lipschitz continuity assumption to provide a bound on the relative distance between the expected risk of the joint model and of a deterministic model used in TRM.
3. **Use of Augmentation Space:** Next, we derive the explicit form of the regularizer $\text{Reg}(\theta)$ in Eq. 5, resorting to the Radon-Nikodym derivative $\frac{dQ}{dP}$, since neither the augmentation prior $P$ nor the augmentation posterior $Q$ are absolutely continuous with respect to the Lebesgue measure.
4. **Optimization of bounds:** Finally, we optimize the bound over its free parameters, which leads to the form of $c_n$.

Observe that if the loss $l(\cdot, y)$ is convex for all $y \in \mathcal{Y}$, then this bound also upper bounds the risk $R_{RM}$ of the aggregate predictors $x \to \mathbb{E}_{g \sim Q_\theta}[h_w(gx)]$.

### 3.3 Building the SCALE Algorithm

We propose to optimize the upper bound from Thm. 3.1 to learn simultaneously the parameters $w$ of the hypothesis space $\mathcal{H} = \{h_w; w \in W\}$ and the parameters $\theta = \{\pi_1, \ldots, \pi_K, \alpha_1, \ldots, \alpha_k\} \in \Theta$ of the distribution $Q_\theta$. We absorb the weight decay term in the optimizer and write the training objective $L(w, \theta)$ as:

$$L(w, \theta) = \frac{1}{n} \sum_{i=1}^{n} \mathbb{E}_{g_1 \sim Q_{(1)}} \left[ \ldots \mathbb{E}_{g_K \sim Q_{(K)}} \left[ l\left( h_w \left( \prod_{j=1}^{K} g_j x_i \right), y_i \right) \right] \right] + \lambda_{\text{reg}} \text{Reg}(\theta = \{\boldsymbol{\pi}, \boldsymbol{\alpha}\}) \tag{6}$$

where $\text{Reg}(\theta = \{\boldsymbol{\pi}, \boldsymbol{\alpha}\})$ is the regularization term suggested by Eq. 5 and $\boldsymbol{\alpha} = [\alpha_1, \ldots, \alpha_k]$, $\boldsymbol{\pi} = [\pi_1, \ldots, \pi_K]$. This regularizer is minimized when each $\pi_i \to 1 - \beta \approx 1$ and each $\alpha_i \to A_i$, increasing the support of the uniform distributions $U_{i, \alpha_i}$. Thus the regularizer promotes frequent augmentations (large $\pi_i$) and a broad range of augmentations (large $\alpha_i$). This leads to a trade-off with the training loss, as desired. During the experiments, we use $\beta = 0.01$.

At test time, we aim to use the learned transformation distribution $Q_\theta$ and model $h_w$ to compute the expected output $\bar{h}_w(x) = \mathbb{E}_{g \sim Q_\theta}[h_w(gx)]$, and we approximate the expectation by Monte Carlo sampling. In the experiments we refer to the use of $\bar{h}_w$ as test-time augmentation.

End to end training requires the computation of the gradients with respect to the local variables $\alpha_i, \pi_i$ that appear in the distributions $Q_{(i)}$ in the iterated expectations of equation 6. The general case of this setting has been adressed in Titsias & Lázaro-Gredilla (2015). In our case, the specific form of the TRM loss allows

---

**Algorithm 1** SCALE: Training for learning augmentations using TRM

---

**Input:** Data $S = \{x_i, y_i\}_{i=1}^n$, $\{h_w\} \in \mathcal{H}$
Initialize $\pi_i = \frac{1}{K}$, $\alpha_i = \varepsilon$, $\theta = \{\boldsymbol{\pi}, \boldsymbol{\alpha}\}$
Initialize model parameters $w = w_0$.
**for** epoch=1 **to** Total Epochs **do**
   **for** batch={batch_x, batch_y} **in** $S$ **do**
     **for** $(x, y)$ **in** batch **do**
       Sample $M$ iid transforms $g^{(j)} \sim Q_\theta$, $j = 1, \ldots, M$, according to equation 4
       Add all $(g^{(j)}x, y)$ to augmented_batch
     **end for**
     Compute batch_output $= h_w($augmented_batch_x$)$
     Compute batch loss $L(w, \theta) = ($batch_y, batch_output$)$ according to equation 6
     Compute $(\widehat{\nabla_\pi L}, \widehat{\nabla_\alpha L})$
     Set parameters $w \leftarrow w - \lambda_w \widehat{\nabla_w L}$, $\boldsymbol{\alpha} \leftarrow \text{clamp}(\boldsymbol{\alpha} - \lambda_\alpha \widehat{\nabla_\alpha L}, \min = 0, \max = A_i)$
     $\boldsymbol{\pi} \leftarrow \text{clamp}(\boldsymbol{\pi} - \lambda_\pi \widehat{\nabla_\pi L}, \min = 0, \max = 1)$
   **end for**
 **end for**

---

us to compute unbiased gradient estimators $\widehat{\nabla_\pi L}$, $\widehat{\nabla_\alpha L}$ with respect to $\boldsymbol{\pi}$, $\boldsymbol{\alpha}$, without the need to resort to generic estimators such as REINFORCE (Williams, 1992). In particular, the form of $Q_{(i)}$ permits us to compute the gradients with respect to the $\pi_i$'s in closed form while we use the reparametrization trick for the $\alpha_i$'s similar to Benton et al. (2020). In Appendix A.3 we derive the unbiased gradient estimators for both parametric and parameter-free base distributions. Algorithm 1 shows the training process for optimizing over the parameters of the augmentation distribution and of the network.

## 4 TRM under Distributional Invariance

In this section, we provide a theoretical result in the setting where the data have some unknown symmetry in the form of a distributional invariance. By drawing a connection between the risks $R_{RM}$ and $R_{TRM}$ from Eq. 1, Eq. 2, we prove that TRM recovers the unknown invariance, as well as a model that is optimal under the "oracle", i.e., the standard risk $R_{RM}$ that knows the invariance.

We denote the respective optimal risks by $R_{RM}^*$ and $R_{TRM}^*$, and define the set $G_\Omega = \bigcup_{Q \in \Omega} G_Q$ of all transforms considered. We also define the hypothesis space of transforms composed with models in $\mathcal{H}$, $\mathcal{H} \circ G_\Omega = \{x \to (h \circ g)(x) \mid h \in \mathcal{H}, g \in G_\Omega\}$.

The first proposition provides a lower bound on the TRM risk when the hypothesis space $\mathcal{H}$ is "large enough" that $\mathcal{H} \circ G_\Omega \subseteq \mathcal{H}$, or that it is closed under $Q$-expectations (so for any $Q \in \Omega$, $h \in \mathcal{H}$ implies $x \to \mathbb{E}_{g \sim Q} h(gx) \in \mathcal{H}$). For detailed proofs we refer to Appendix A.1.:

**Proposition 4.1** (Lower Bound). *For any $h \in \mathcal{H}$ and $Q \in \Omega$, we have $R_{TRM}(h, Q) \geq R_{RM}^*$ under either one of the following two sufficient conditions:*

    *1. If $\mathcal{H} \circ G_\Omega \subseteq \mathcal{H}$; or*

    *2. if $l(\cdot, y)$ convex for all $y \in \mathcal{Y}$, and $\mathcal{H}$ is closed under $Q$-expectations for all $Q \in \Omega$.*

Condition 1 is not restrictive, because model spaces of interest are typically very large and expressive. For example if $\Omega$ contains distributions on 2D affine transformations, then even the simple model class $\mathcal{H}$ of all linear classifiers satisfies the condition.

The next proposition shows how an assumption about the type of augmentations within $G_\Omega$ can lead to an upper bound on the risk $R_{TRM}^*$.

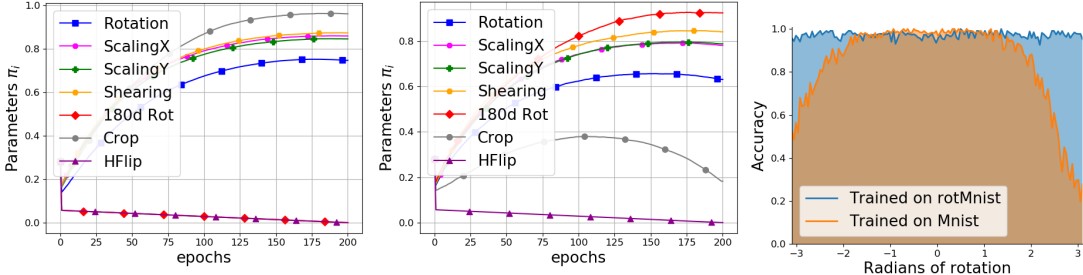

Figure 3: Parameters $\pi_i$ during training using SCALE on **Left**: MNIST, **Center**: rotMNIST. **Right**: Accuracy of models trained on MNIST and rotMNIST, using MNIST images rotated by various angles.

**Proposition 4.2** (Upper Bound). *Let $G_{inv} = \{g \in G_\Omega : (X, Y) =_d (gX, Y)\}$ be the set of distributional invariances $g$. If there is a distribution $Q \in \Omega$ such that its support $G_Q$ is included in the set of distributional invariances, i.e., $G_Q \subseteq G_{inv}$, then $R^*_{TRM} \leq R^*_{RM}$.*

When the conditions for both the upper and lower bounds hold, then the optimal classifier under the TRM risk is also optimal for the standard risk $R_{RM}$. We let $h^*_{RM}$ be any optimal classifier under the standard risk $R_{RM}$. Further, a classifier $h$ is $G$-invariant if for all $g \in G$ and all $x \in \mathcal{X}$, $h(gx) = h(x)$.

**Corollary 4.3** (Zero gap under distributional invariance).

1. *Assume $\mathcal{H} \circ G_\Omega \subseteq \mathcal{H}$ and that there is a $Q \in \Omega$ such that $G_Q \subseteq G_{inv}$. Then, $R^*_{TRM} = R^*_{RM}$ and any pair $(h^*_{RM}, Q)$ minimizes the TRM risk.*

2. *If in addition the conditions in Proposition 4.1, part 2, hold, and there is a compact group $G \subseteq G_{inv}$ such that $\mathrm{Unif}(G) \in \Omega$[8], then there exists a $G$-invariant TRM-optimal classifier $h^*_{TRM}$.*

This corollary is consistent with the experimental results in Section 5.2 for RotMNIST—an approximately rotationally invariant dataset—and serves as a proof of concept for the functionality of our SCALE algorithm.

## 5 Experiments

### 5.1 Base Transformations

We parametrize $Q_\theta$ using a set of base transformations commonly used in image processing. Specifically, in both experiments presented in Sections 5.2, 5.3 we learn a distribution $Q_\theta$ where each sample $g \sim Q_\theta$ is constructed as $g = g_1 g_2 \ldots g_K$, with $K = 7$.

The first four transformations $g_1, \ldots, g_4$ correspond to rotation, $X$-axis scaling, $Y$-axis scaling and $X$-axis shearing. Each transformation is applied with probability $\pi_i$, and its parameter is sampled uniformly from $[-\alpha_i, \alpha_i]$. The transformations $g_5$, $g_6$ correspond to discrete rotations by 180 degrees, and horizontal flips, respectively. They are applied with probabilities $\pi_5$, $\pi_6$, respectively, and once they are applied, they perform the discrete rotation or horizontal flip with probability 0.5. Finally $g_7$ is a random cropping of the image, applied with probability $\pi_7$. (For more details see Appendix A.5.)

### 5.2 Experiments on MNIST, rotMNIST: Discovering Distributional Invariances

Using the parametrization from Section 5.1, we train on MNIST (LeCun et al., 1998) and rotated MNIST (rotMNIST). We train a simple CNN similar to the model used in Benton et al. (2020). Figure 3 shows how the probabilities $\pi_i$ of the transformation distributions change during the training.

For MNIST, the parameters $\pi_i$ giving the probabilities of rotation, scaling, shearing and random cropping increase and converge to values above 0.6, while the $\pi_i$-s for $180^o$ rotation and horizontal flips converge to

---

[8]The theorem holds mutatis mutandis for locally compact groups too, if we extend $\Omega$ to contain more general (not just probability) measures, e.g., the right-invariant Haar measure $U_G$; and replace expectations with integrals.

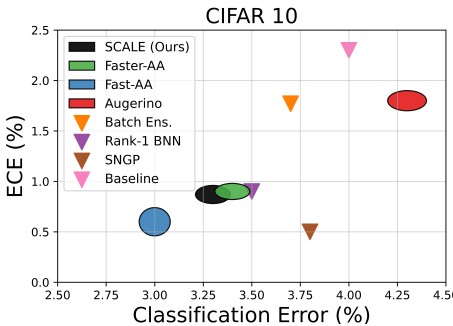 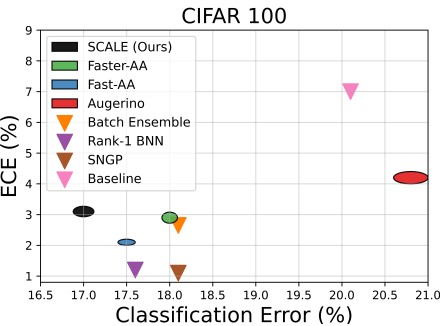

Figure 4: Comparison of the ECE and Classification Error achieved by the various methods on CIFAR10/100. In the methods where test-time augmentation was performed the standard deviation in the results is indicated by the width and the height of the ellipses (corresponding to two standard deviations). For the methods drawn with a triangle the standard deviation is not provided.

zero. For the rotation, the learned distribution corresponds to angles uniformly sampled from $[-0.31, 0.31]$ radians, which is consistent with the common practice of augmenting with small rotations.

When training on rotMNIST, we observe that the parameters $\pi_1, \ldots, \pi_5$ for rotation (continuous and discrete), scaling and shearing converge to values above 0.6, while the parameters $\pi_6, \pi_7$ for horizontal flips and random crops go close to zero. This indicates that on rotMNIST, SCALE selects augmentations that are useful on the original dataset (MNIST) and also all rotations that correspond to the invariance of the dataset. Additionally, the right plot of Figure 3 shows that—contrary to MNIST—on rotMNIST SCALE results in an invariant classifier. These results are consistent with Corollary 4.3. On rotMNIST we achieve test accuracy of 99.1%, which compares favorably to Augerino, that achieves test accuracy of 98.9% on the same network. According to Benton et al. (2020) only Weiler & Cesa (2019) have reported a better result using a network equivariant to rotations.

### 5.3 Learning Data Augmentations on CIFAR 10/ 100

Using SCALE and the parametrization shown in Section 5.1, we learn useful data augmentations on the CIFAR 10 and CIFAR 100 datasets (Krizhevsky, 2009). For both CIFAR 10 and CIFAR 100 we used a Wide ResNet 28-10 (Zagoruyko & Komodakis, 2016) (for training details see Appendix A.4).

We compare the results for our method and for prior works on learning data augmentations that are discussed in Section 6. For Trivial-Augment (TA) (Müller & Hutter, 2021) and DADA (Li et al., 2020) we evaluate using the same augmentation space as our method (presented in Section 5.1 and Appendix A.5). First, we evaluate the trained models on the original test images, without applying any test-time augmentation. Additionally, for each method, we perform test-time augmentation by sampling augmented copies from the augmentation policy learned by the corresponding method and taking the average prediction. Table 1 shows the accuracy for CIFAR 10 and CIFAR 100, both using test-time augmentation with 4 and 8 transforms, and without test-time augmentation. Finally Table 2 shows an ablation study on the use of our proposed parametrization and regularizer.

On both CIFAR 10 and CIFAR 100, SCALE converges to random rotations, scalings, shearings, horizontal flips and crops, while it does not pick $180^o$ discrete rotations. SCALE outperforms Augerino and DADA on both CIFAR 10 and CIFAR 100. Also, SCALE performs similarly to Fast-AutoAugment Lim et al. (2019) (Fast-AA) and Faster-AutoAugment Hataya et al. (2020) (Faster-AA) —that first search for an augmentation policy and then train a model—on CIFAR 10 and outperforms both methods on CIFAR 100.

### 5.4 Learning Augmentations can Improve Calibration

In addition to accuracy, many applications require a model to be well calibrated, which means that it provides an accurate uncertainty estimate for its predictions. In a classification setting we can use the output of the softmax layer as an indication of the model's confidence that the input belongs to a specific

Table 1: Test accuracy of Wide ResNet 28-10 trained on CIFAR 10/100, using several method for learning data augmentations. We show the test accuracy of the models when we do not perform test-time augmentation (No TestAug.) and when we perform test-time augmentation (TestAug.) with $n = 4, 8$ samples. The values presented are the average over five trials. For all methods, the standard deviation is close to 0.1% (please see the appendix for a table containing the standard deviations) We also report the additional time that each method spends in policy search in GPU Hours relative to the standard training (No Augm. column)

| | | No Augm. | Augerino +Flips | Fast-AA | Faster-AA | TA | DADA (geometric) | **SCALE (Ours)** |
|---|---|---|---|---|---|---|---|---|
| CIFAR10 | No TestAug. | 92.2% | 95.7% | **97.2**% | 97.1% | 97.2% | 96.6% | 96.7% |
| | TestAug. (N=4) | - | 95.7% | **97.0**% | 96.6% | 96.8% | 96.5% | 96.7% |
| | TestAug. (N=8) | - | 95.8% | **97.2**% | 96.8% | 96.9% | 96.8% | 96.9% |
| CIFAR100 | No TestAug. | 73.1% | 79.4% | 82.5 % | 81.9% | **83.1**% | 79.4% | 82.7% |
| | TestAug. (N=4) | - | 79.2% | 82.5 % | 82.0% | 82.9% | 79.8% | **83.0**% |
| | TestAug. (N=8) | - | 79.3% | 82.9 % | 82.4% | **83.6**% | 80.1% | 83.3% |
| GPU Hours (CIFAR 10/100) | | 5h | +1h | +8h | +3h | +1h | +2.5h | +1.5h |

Table 2: Ablation study on the proposed regularization and parametrization. First we evaluate a model using the Augerino parametrization and our proposed PAC-Bayes regularizer (without the addition of flips). Second we evaluate a model using the SCALE parametrization and the regularizer used in Augerino ($L_2$ regularizer). Finally we show our method with both our proposed regularizer and parametrization

| | | Augerino (No Flips) + PB regularizer | SCALE + Aug regularizer | **SCALE (Ours)** |
|---|---|---|---|---|
| CIFAR10 | No TestAug. | 94.2% | 96.6% | **96.7**% |
| | TestAug. (N=4) | 94.3% | 96.7% | **96.7**% |
| | TestAug (N=8) | 94.4% | 96.8% | **96.9**% |

class. In particular, consider a model $h : \mathcal{X} \to \mathbb{R}^{|\mathcal{Y}|}$ that assigns score $h_y$ (say, a softmax output) to each class $y$. As discussed in (Cox, 1958; DeGroot & Fienberg, 1983; Gneiting et al., 2007; Guo et al., 2017), given a random datapoint $(X, Y)$ drawn from the data distribution $\mathcal{D}$, a model $h$ is perfectly calibrated if $\mathbb{P}_{(X,Y)\sim\mathcal{D}}(Y = y|\ h_y(X) = p) = p, \quad \forall p \in [0, 1]$.

A commonly used metric for quantifying the calibration of a model is the Expected Calibration Error (ECE) (see e.g., (Harrell, 2015; Naeini et al., 2015)): the samples are partitioned into $M > 0$ equally spaced bins according to the confidence $h_{\hat{y}}(x)$ for the predicted category $\hat{y} = \text{argmax}_y(h_y(x))$. Let $B_m$ be the set containing the indices of datapoints that belong to the $m$-th bin, for $m \in [M]$. For the $m$-th bin we define the mean accuracy as $\textbf{acc}(B_m) = \frac{1}{|B_m|}\sum_{i\in B_m}\mathbb{1}(\hat{y}_i = y_i)$ and the mean confidence as $\textbf{conf}(B_m) = \frac{1}{|B_m|}\sum_{i\in B_m}h_{\hat{y}_i}(x_i)$. We can then define the Expected Calibration Error as: $\text{ECE} = \sum_{m=1}^{M}|\textbf{acc}(B_m) - \textbf{conf}(B_m)| \cdot |B_m|/n$.

We compare a baseline deterministic model trained with the standard augmentations presented in Zagoruyko & Komodakis (2016) and evaluated without test-time augmentation to models trained with SCALE, Augerino, Fast-AA and Faster-AA and evaluated with test-time augmentation using 4 augmented datapoints Gawlikowski et al. (2021). Additionally we compare with methods designed to improve the calibration of a model that are presented in Nado et al. (2021). Specifically we compared with Batch Ensemble Wen et al. (2020), Rank-1 BNN Dusenberry et al. (2020) and SNGP Ensemble Liu et al. (2020). For the latter three methods we use the implementation and results provided in Nado et al. (2021). For both CIFAR 10 and CIFAR 100, all methods use the same WideResNet 28-10 network architecture.

Figure 4 compares the calibration and classification errors achieved by the various methods on CIFAR 10/100. SCALE outperforms the baseline and Augerino, and obtains classification error and ECE comparable to more computationally expensive methods that learn augmentation policies (Faster-AA, Fast-AA), and also to methods that aim to improve the calibration of models. It is important to note that SCALE mainly aims to minimize the classification error of a model, and not its calibration error. Models that only optimize the classification performance need not be well calibrated. Thus, perhaps surprisingly, we show that SCALE results in models not only with low classification error (on CIFAR 100 it achieves the lowest error) but also

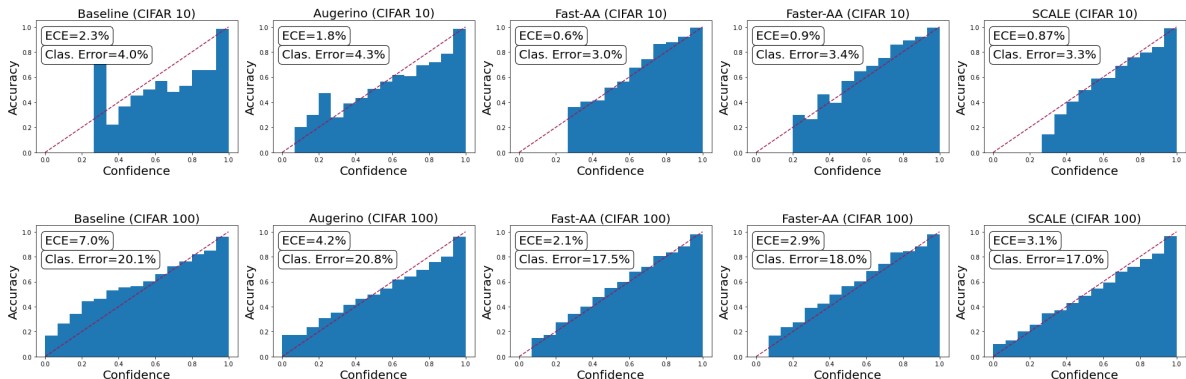

Figure 5: Reliability diagrams of various methods for learning data augmentations on CIFAR 10/100.

low with calibration error—close to the error achieved by methods that mainly aim to improve calibration. We also present reliability diagrams that show, for each confidence bin, the mean accuracy achieved by the corresponding method. Figure 5 shows the reliability diagrams of the baseline deterministic model and of the models trained with SCALE, Augerino, Fast-AutoAugment and Faster-AutoAugment which are evaluated using test-time augmentation with 4 augmented transforms.

### 5.5 Learning Data Augmentations on Language Navigation Task

We showcase the use of our method on a recently proposed task of language navigation, where the set of best augmentations is not generally known. Specifically in this task, given a semantic map and a natural language instruction, we aim to infer a sequence of waypoints that define a path that follows the given instruction. For our experiment we follow the definition of the task proposed in Krantz et al. (2020), where the maps are extracted from scenes of the Matterport3D dataset (Chang et al., 2017) and the instructions are given from the R2R dataset (Anderson et al., 2018).

We apply SCALE over the path prediction model shown in Figure 6b, which was proposed in Georgakis et al. (2022). We can measure the performance of the predicted waypoints by comparing them to the waypoints of the ground truth path. We consider a predicted waypoint as correct if it is located at most one meter away from the ground truth. We focus on learning transformations applied on the input semantic map using the parametrization described in Section 5.1. In the figure 6a, we show the performance achieved by our method compared to the performance of the baseline model as a function of the regularization parameter $\lambda_{reg}$. We can observe that for values of $\lambda_{reg}$ ranging from $5*10^{-4}$ to $3.5*10^{-3}$ our method learns augmentations that result in increased performance compared to the performance achieved by the baseline.

## 6 Related Work

There is a vast amount of related work; we can only discuss the most closely related ones.

**VRM.** We can view ETRM as a generalization of Vicinal Risk Minimization (VRM) (Chapelle et al., 2001). In VRM the Dirac masses around each training datapoint are replaced with vicinal functions, usually uniform or Gaussian distributions centered at each point. It has been proven empirically (Cao & Rockett, 2015) and theoretically (Zhang et al., 2018) that VRM enjoys some benefits over ERM, but is highly dependent on the right choice of the vicinal functions. Instead of fixed vicinal functions, we learn transformation distributions. In this sense, our learning algorithm adapts the vicinal functions to the task.

**Invariance.** Data augmentation can be viewed as an implicit restriction to models invariant to predefined transformations. When the invariance is known, and the transformations have a group structure, an exact way to guarantee the invariance is to "bake it in" the network design. This can be traced back to (Fukushima, 1980; LeCun et al., 1989) with the development of CNNs, which use convolutions to build translation equivariant layers. Architectures that extend the idea to discrete (Cohen & Welling, 2016) or continuous (Weiler et al., 2018) rotational invariance and to more general Lie Groups (Finzi et al., 2020) have been proposed. More recently they have been generalized beyond Euclidean domains (Esteves et al., 2018; Maron et al., 2019; Cohen et al., 2019). When the data symmetry is unknown, Anselmi et al. (2019) assume that the

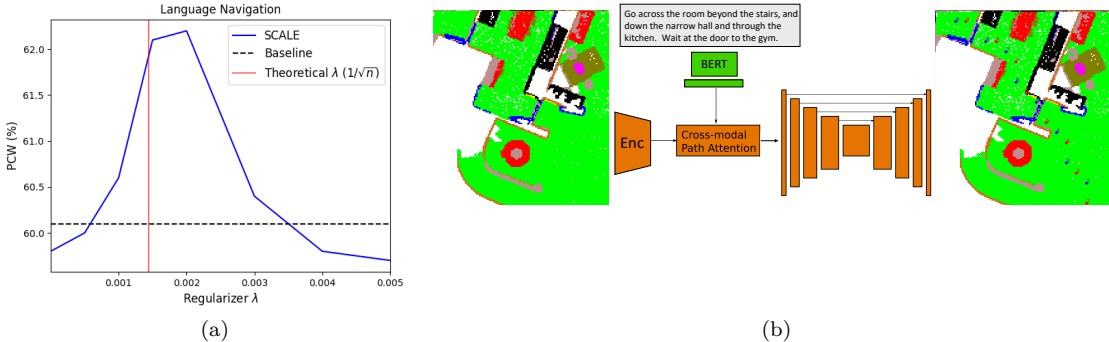

Figure 6: (a) Percentage of correct waypoints (PCW) predicted by the language navigation model when we train only the original model (Baseline) and when we incorporate SCALE into the training of the model with different regularization parameters $\lambda_{reg}$. (b) A high level overview of the language navigation model used in Georgakis et al. (2022), which is given a semantic map and an instruction and outputs ten waypoint predictions (red dots) that aim to match the ten waypoints of the ground truth path (blue dots).

dataset is partitioned into full orbits of an unknown group and propose a symmetry-adapted regularization to find it.

**Data augmentation (DA).** The concept of data augmentation in machine learning goes back at least to Baird (1992). More recently, generative networks have been used to generate augmented data (Mirza & Osindero, 2014; Antoniou et al., 2018; Robey et al., 2020). There is a variety of work on designing augmentations (Zhong et al., 2017; Lopes et al., 2019; Perlin, 1985; Zhang et al., 2017). Vedaldi et al. (2011) propose a method for learning equivariant SVM regressors. Novotny et al. (2018) introduce augmentations on the fly to learn equivariant descriptors in a self-supervised setting. Deeper theoretical understanding on the relative efficiency gains of DA has been obtained by Chen et al. (2020), who develop a group-theoretic framework, and Lyle et al. (2019), who compare feature averaging with data augmentation on the grounds of generalization using PAC-Bayes theory under the assumption that $(X, Y) =_d (gX, Y)$. We derive the bound jointly over $(Q, h)$ without this assumption.

Ratner et al. (2017) use a generative adversarial approach to learn sequences of predefined valid transformations. Similarly AutoAugment (Cubuk et al., 2019a) learns augmentation policies using reinforcement learning to search over a predefined space of transformations. Many works aim to reduce the high computational complexity of the AutoAugment policy search: RandAugment (Cubuk et al., 2019b) simplifies the search space to two hyperparameters, Fast-AutoAugment (Lim et al., 2019) proposes a more efficient search strategy based on density matching and Faster-AutoAugment (Hataya et al., 2020) introduces a gradient-based search method that minimizes the distance between the original and the augmented dataset. Similarly to our method, Lin et al. (2019) propose to formulate the augmentations as parametrized distributions. They suggest a bi-level optimization algorithm to train the model simultaneously with the augmentation parameters using policy gradients. While faster than AutoAugment this approach still introduces a very big overhead over standard training. Hataya et al. (2022) also propose a bi-level optimization problem to directly optimize the validation accuracy without proxy tasks or smaller augmentation spaces by approximating the inverse Hessian via a truncated von Neumann series, an approximation that was first studied in Lorraine et al. (2019). A similar automatic augmentation algorithm was proposed in Raghu et al. (2022) for electro-cardiograms. Finally Müller & Hutter (2021) propose Trivial-Augment, a simple tuning-free augmentation policy, and argue it achieves comparable or better performance than the above methods presented.

van der Wilk et al. (2018) cast the problem as Bayesian model selection and learns data transformations jointly with a downstream task by maximizing the marginal likelihood. This is applied to Guassian processes by imposing an invariance constraint on the kernel. The method is restricted to differentiable transformations and not directly applicable to deep neural networks due to the inefficiency to scale the marginal likelihood. Extending this work to invariant deep kernel learning Schwöbel et al. (2022) propose to compute the marginal likelihood only in the last layer of a deep network and show some performance gains but also report scalability issues. Lastly, Immer et al. (2022) extend this formalism to deep networks by proposing a Laplace

approximation to the marginal likelihood. While they demonstrate some performance gain the computation that involves gradients of the log determinant of the Hessian of the model introduces big overhead over standard training.

Augerino (Benton et al., 2020) optimizes over a continuous parametrization of augmentation distributions, and thus excludes discrete transformations (e.g., horizontal flips). Augerino uses the negative norm of the parameters of the learned augmentation distribution as a regularization term. This choice of regularizer may be more unstable, because it leads to minimizing a concave function. We compare favorably with Augerino in both accuracy and calibration in the experimental section. Similarly to our method, DADA (Li et al., 2020) relaxes the discrete data augmentation policy selection to a differentiable optimization problem. They rely on the Gumbel-Softmax reparametrization trick introduced in Maddison et al. (2017); Jang et al. (2017) and apply variance reduction to their proposed gradient estimator. We compare favorably against DADA in all CIFAR 10/100 and SVHN.

## 7 Conclusion

In this paper, we formulated the Transformed Risk Minimization framework, and used it to design SCALE, a new algorithm for learning augmentation distributions. This relies on a new parametrization of the augmentation space via a composition of stochastic blocks, and leverages PAC-Bayes theory to derive a novel training objective. We provided experimental results that corroborate our theoretical analysis, and showed that SCALE can lead to advantages over prior approaches in both accuracy and calibration and can be incorporated easily in large-scale problems like language navigation and benefit them in terms of performance.

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

## A   Appendix

### A.1   TRM Propositions

For technical reasons, in the definitions of optimal risks, we switch from minimization to taking infima. This makes the definitions more broadly applicable. When the minima are well-defined, we recover the definitions from the main body. We also assume that all functions we consider are measurable, including after taking the appropriate infimum over the hypothesis space in the definitions of optimal risks.

In this section we provide the proofs of the propositions in Section 4. The following definitions will be used extensively in the proofs:
RM risk and its optimal value:

$$R(h) := \mathbb{E}_{(X,Y)\sim\mathcal{D}}[l(h(X),Y)], \qquad R^*_{RM} := \inf_{h\in\mathcal{H}} R(h).$$

TRM risk and its optimal value:

$$R(h,Q) := \mathbb{E}_{(X,Y)\sim\mathcal{D}}\mathbb{E}_{g\sim Q}[l(h(gX),Y)], \qquad R^*_{TRM} := \inf_{h\in\mathcal{H},Q\in\Omega} R(h,Q).$$

*Fixed g-risk* for a fixed transformation $g$:

$$R_f(h,g) := \mathbb{E}_{(X,Y)\sim\mathcal{D}}[l(h(gX),Y)].$$

Set of *distributional invariances* $g$:

$$G_{inv} := \{g \in G_\Omega : (X,Y) =_d (gX,Y)\}. \tag{7}$$

**Proof of Proposition 4.1 (Lower Bound).**   First we prove the following lemma:

**Lemma A.1.**   *For every distribution $Q$,*

$$\inf_{h\in\mathcal{H}} \mathbb{E}_{g\sim Q} R_f(h,g) \geq \mathbb{E}_{g\sim Q} \inf_{h\in\mathcal{H}} R_f(h,g).$$

*Proof of Lemma A.1.*   Using the definition of the infimum and the monotonicity of expectations

$$R_f(h,g) \geq \inf_{h\in\mathcal{H}} R_f(h,g), \;\; \forall h \in \mathcal{H} \implies$$

$$\mathbb{E}_{g\sim Q} R_f(h,g) \geq \mathbb{E}_{g\sim Q} \inf_{h\in\mathcal{H}} R_f(h,g), \;\; \forall h \in \mathcal{H} \implies$$

$$\inf_{h\in\mathcal{H}} \mathbb{E}_{g\sim Q} R_f(h,g) \geq \mathbb{E}_{g\sim Q} \inf_{h\in\mathcal{H}} R_f(h,g).$$

$\square$

Next, for any fixed $g$ we have:

$$R_f(h,g) = R(h \circ g) \implies \inf_{h\in\mathcal{H}} R_f(h,g) = \inf_{\tilde{h}\in\mathcal{H}\circ g} R(\tilde{h}), \tag{8}$$

where $\mathcal{H} \circ g := \{h \circ g | h \in \mathcal{H}\}$.

Using the lemma and observation in equation 8, we now complete the proof of the proposition.

*Proof of Proposition 4.1.*   We prove the two claims in turn.

1. It suffices to show that $R^*_{TRM} \geq R^*_{RM}$. For this, we have

$$R^*_{TRM} = \inf_{h\in\mathcal{H},Q\in\Omega} R(h,Q) = \inf_{Q\in\Omega} \inf_{h\in\mathcal{H}} R(h,Q)$$

$$= \inf_{Q\in\Omega} \inf_{h\in\mathcal{H}} \mathbb{E}_{g\sim Q} \mathbb{E}_{(X,Y)\sim\mathcal{D}}[l(h(gX),Y)] = \inf_{Q\in\Omega} \inf_{h\in\mathcal{H}} \mathbb{E}_{g\sim Q} R_f(h,g),$$

where in the first line we used that the joint infimum is the same as the sequential infimum. In the second line we used the Fubini-Tonelli theorem to change the order of expectations, along with the definition of fixed $g$-risk. By Lemma A.1, the last expression is lower bounded by

$$\inf_{Q \in \Omega} \mathbb{E}_{g \sim Q} \inf_{h \in \mathcal{H}} R_f(h, g) = \inf_{Q \in \Omega} \mathbb{E}_{g \sim Q} \inf_{\tilde{h} \in \mathcal{H} \circ g} R(\tilde{h}) \geq \inf_{Q \in \Omega} \mathbb{E}_{g \sim Q} R_{RM}^* = R_{RM}^*.$$

For the first equation, we used the observation in equation 8, and in the following inequality we used that for any fixed $g \in G_\Omega$ we have $\mathcal{H} \circ g \subseteq \mathcal{H} \circ G_\Omega \subseteq \mathcal{H}$ from the assumption of the proposition.

2. We have that if $h \in \mathcal{H}$ then $\bar{h}_Q(x) := \mathbb{E}_{g \sim Q} h(gx) \in \mathcal{H}$ for any $Q \in \Omega$. Since the loss $l : \mathcal{Y}' \times \mathcal{Y} \to \mathbb{R}$ is convex for all $y \in \mathcal{Y}$ we have from Jensen's inequality that for all $h \in \mathcal{H}, Q \in \Omega$:

$$\begin{aligned} R(h, Q) = \mathbb{E}_{(X,Y) \sim \mathcal{D}} \mathbb{E}_{g \sim Q}[l(h(gX), Y)] &\geq \mathbb{E}_{(X,Y) \sim \mathcal{D}}[l(\mathbb{E}_{g \sim Q} h(gX), Y)] \\ &= \mathbb{E}_{(X,Y) \sim \mathcal{D}}[l(\bar{h}_Q(X), Y)] = R(\bar{h}_Q) \\ &\geq \inf_{h \in \mathcal{H}} R(h) = R_{RM}^*. \end{aligned} \tag{9}$$

$\square$

**Discussion.** The condition $\mathcal{H} \circ \text{supp}(\Omega) \subseteq \mathcal{H}$ means that for each transformation $g \in G$ used by any distribution in $\Omega$, and each $h \in \mathcal{H}$, we can write $h(gx) = h'(x)$ for some $h' \in \mathcal{H}$. Thus, this means that minimizing the average risk over $\mathcal{H} \circ \text{supp}(\Omega)$ is a type of constraint compared to minimizing the risk over all hypotheses in $\mathcal{H}$.

Data augmentation is not usually viewed as imposing a constraint. The goal is to learn an invariant predictor, but the motivation for data augmentation is that we do not know how to compute one, so we use a heuristic way to expand our dataset. Thus, in this sense, averaging the risk over transforms can be viewed as a form of a constraint.

**Proof of Proposition 4.2 (Upper Bound).**

*Proof.* Suppose $Q_{inv}$ is such that $Q_{inv} \in \Omega$, and $G_{Q_{inv}} \subseteq G_{inv}$. Then,

$$\begin{aligned} R(h, Q_{inv}) = \mathbb{E}_{(X,Y) \sim \mathcal{D}} \mathbb{E}_{g \sim Q_{inv}}[l(h(gX), Y)] &= \mathbb{E}_{g \sim Q_{inv}} \mathbb{E}_{(X,Y) \sim \mathcal{D}}[l(h(gX), Y)] \\ &= \mathbb{E}_{g \sim Q_{inv}} \mathbb{E}_{(X,Y) \sim \mathcal{D}}[l(h(X), Y)] = R(h), \end{aligned} \tag{10}$$

where we used the Fubini-Tonelli theorem to change the order of expectations, and then used that $(X, Y) =_d (gX, Y)$, for all $g \in G_{Q_{inv}}$ to conclude that $\mathbb{E}_{(X,Y) \sim \mathcal{D}}[l(h(gX), Y)] = \mathbb{E}_{(X,Y) \sim \mathcal{D}}[l(h(X), Y)]$. Therefore,

$$R_{TRM}^* = \inf_{h \in \mathcal{H}, Q \in \Omega} R(h, Q) \leq \inf_{h \in \mathcal{H}} R(h, Q_{inv}) = \inf_{h \in \mathcal{H}} R(h) = R_{RM}^*.$$

This finishes the proof. $\square$

**Discussion.** The upper bound uses that we include the "correct" invariances in a distribution $Q \in \Omega$, so the risk does not increase. This seems to be a generally justified setting for TRM. The data distribution is invariant under some transforms in $G$, so we use this knowledge to minimize the empirical transformed risk, a better estimator of the test error than the simple empirical risk. From arguments in Chen et al. (2020), it follows that for any fixed $Q$ supported on $G_{inv}$, the empirical TR has the same mean, but a smaller variance than the empirical risk. Clearly, the same still holds uniformly over all such $Q \in \Omega$, which provides a justification that the TRM framework provides a uniformly more accurate estimator of the population error than RM.

**Proof of Corollary 4.3 (Zero Gap).**

*Proof.*

1. From Proposition 4.1 (Part 1), using the assumption $\mathcal{H} \circ G_\Omega \subseteq \mathcal{H}$, we obtain the lower bound $R^*_{TRM} \geq R^*_{RM}$. Since there is a distribution $Q_{inv} \in \Omega$ such that $G_{Q_{inv}} \subseteq G_{inv}$, from Proposition 4.2 we obtain the upper bound $R^*_{TRM} \leq R^*_{RM}$. Thus, $R^*_{TRM} = R^*_{RM}$.

   Assume now that $h^*_{RM}$ is a minimizer of the risk $R(h)$ over $\mathcal{H}$. Then from equation 10, $R(h^*_{RM}, Q_{inv}) = R(h^*_{RM}) = R^*_{RM} = R^*_{TRM}$. So, the pair $(h^*_{RM}, Q_{inv})$ is a minimizer of the TRM risk.

2. Set $U := \mathrm{U}(G)$. From part 1 above, if $h^*_{RM}$ is a minimizer of the standard risk $R(h)$ over $\mathcal{H}$, then $(h^*_{RM}, U)$ is a minimizer of the TRM risk $R(h, Q)$ over $\mathcal{H} \times \Omega$, since $G \subseteq G_{inv}$ and $U \in \Omega$ from the assumptions of the theorem.

   From equation 9 we have that $R^*_{RM} = R^*_{TRM} = R(h^*_{RM}, U) \geq R(\bar{h}^*_{RM,U})$, where $\bar{h}^*_{RM,U}(x) = \mathbb{E}_{g \sim U} h^*_{RM}(gx) \in \mathcal{H}$. Hence, $\bar{h}^*_{RM,U}$ is also a minimizer of the RM risk over $\mathcal{H}$ and the pair $(\bar{h}^*_{RM,U}, U)$ is a minimizer of the TRM risk over $\mathcal{H} \times \Omega$.

   Now we observe that $\bar{h}^*_{RM,U}$ is $G$-invariant, since for all $g$ in the group $G$ we have:

$$\bar{h}^*_{RM,U}(gx) = \mathbb{E}_{g' \sim U} h^*_{RM}(g'gx) = \int_{g' \in G} h^*_{RM}(g'gx) dU(g')$$

$$= \int_{\tilde{g}g^{-1} \in G} h^*_{RM}(\tilde{g}x) dU(\tilde{g}g^{-1}) = \int_{\tilde{g} \in G} h^*_{RM}(\tilde{g}x) dU(\tilde{g})$$

$$= \mathbb{E}_{\tilde{g} \sim U} h^*_{RM}(\tilde{g}x) = \bar{h}^*_{RM,U}(x).$$

   Above we changed variables to $\tilde{g} = g'g$, used that the group is "closed", i.e., for any fixed $g \in G$, $\tilde{g}g^{-1} \in G \iff \tilde{g} \in Gg = G$, and that $dU(\tilde{g}g^{-1}) = dU(\tilde{g})$, since the Haar measure is right-invariant.

$\square$

**Discussion.** This "zero gap" result follows by putting together the upper and lower bounds. It requires that we include the correct invariances in a distribution $Q$, but that the transformations in this distribution do not go outside the hypothesis space $\mathcal{H}$. This is an ideal setting, in the sense that we use invariances as "inductive biases" to obtain a better estimator of the population error, but the hypothesis space is rich enough that all the transforms we use can be realized by actual hypotheses.

In addition, the result shows that if the loss is convex, then the optimal classifier is invariant to $G$, but of course $\mathcal{H}$ needs to contain this classifier. In that case, there are many optimization objectives that have that hypothesis as a minimizer, in particular RM and TRM. However, TRM has the advantages described before: (1) reduced variance, and (2) fast inference.

### A.2 Proof of PAC-Bayes bound (Theorem 3.1)

We will break the proof into three parts. We start by discussing the preliminary theorems from the PAC-Bayes literature that we will use to derive our bound. We then use the assumptions of Lipschitz continuity from the theorem to prove some key lemmas. Finally, we derive the bound on TRM and compute the KL divergence terms for the parametrization of SCALE.

#### A.2.1 Preliminaries

Consider the hypothesis space $\mathcal{H}$ parametrized on some space $W \subseteq \mathbb{R}^p$ for some $p \in \mathbb{N}$. Also consider $R_{RM}$, the standard risk from equation 1, and its empirical counterpart $R_{RM,n}(h_w) = \frac{1}{n} \sum_{i=1}^n l(h_w(X_i), Y_i)$, and denote $S = \{X_i, Y_i\}_{i=1}^n \sim \mathcal{D}^n$. Assume $\pi$ is a prior probability measure on $W$. Then, we have the following definition due to Alquier et al. (2016):

**Definition A.2** (Hoeffding assumption)**.** *The prior $\pi$ satisfies the Hoeffding assumption if there exists a function $f$ and a nonempty open interval $I \subset \mathbb{R}_+^*$ such that for any $\lambda \in I$ and for any $w \in W$:*

$$\mathbb{E}_{w \sim \pi} \mathbb{E}_{(X,Y)^n \sim \mathcal{D}^n} \exp\{\pm\lambda(R_{RM}(h_w) - R_{RM,n}(h_w))\} \leq \exp\{f(\lambda, n)\}.$$

Using this assumption, Alquier et al. (2016) prove the following theorem:

**Theorem A.3** (Theorem 4.1 (Alquier et al., 2016))**.** *Given any distribution $\pi$ over $W$ satisfying the Hoeffding assumption, and for any $\delta \in (0, 1]$, $\lambda > 0$, with probability at least $1 - \delta$ over the draw of $S = \{X_i, Y_i\}_{i=1}^n \sim \mathcal{D}^n$, we have simultaneously for all probability measures $\rho \ll \pi$ on $W$ that:*

$$\mathbb{E}_{w \sim \rho} R_{RM}(h_w) \leq \mathbb{E}_{w \sim \rho} R_{RM,n}(h_w) + \frac{1}{\lambda}\left(KL(\rho||\pi) + \log\left(\frac{1}{\delta}\right) + f(\lambda, n)\right).$$

In particular, the Hoeffding assumption above holds for *sub-Gaussian* losses. The following definition is due to Germain et al. (2016a).

**Definition A.4.** *A loss function $l : \mathcal{Y}' \times \mathcal{Y} \to \mathbb{R}$ is sub-Gaussian with variance factor $s^2$ under a prior $\pi$ and a data distribution $\mathcal{D}$ if $V = R_{RM}(h_w) - l(h_w(X), Y)$ is a $s^2$-sub-Gaussian random variable, i.e.,*

$$\ln \mathbb{E}_{w \sim \pi} \mathbb{E}_{(X,Y) \sim \mathcal{D}} \exp(\lambda V) \leq \frac{\lambda^2 s^2}{2}, \forall \lambda \in \mathbb{R}.$$

The following lemma is derived from Theorem A.3 above:

**Lemma A.5** (Germain et al. (2016a))**.** *Given any distribution $\pi$ over $W$, given $\delta \in (0, 1]$, $\lambda > 0$, if the loss $l$ is sub-Gaussian with variance factor $s^2$, with probability at least $1 - \delta$ over the draw of $S = \{X_i, Y_i\}_{i=1}^n \sim \mathcal{D}^n$, we have simultaneously for all probability measures $\rho \ll \pi$ on $W$ that:*

$$\mathbb{E}_{w \sim \rho} R_{RM}(h_w) \leq \mathbb{E}_{w \sim \rho} R_{RM,n}(h_w) + \frac{1}{\lambda}\left(KL(\rho||\pi) + \log\left(\frac{1}{\delta}\right) + \frac{\lambda^2 s^2}{2n}\right).$$

In particular, the authors study the case that the loss $l$ is bounded in $[a, b]$. More specifically, Hoeffding bound implies that $l : \mathcal{Y}' \times \mathcal{Y} \to [a, b]$ is a sub-Gaussian loss with variance factor $s^2 = (b-a)^2$ and the bound becomes:

$$\mathbb{E}_{w \sim \rho} R_{RM}(h_w) \leq \mathbb{E}_{w \sim \rho} R_{RM,n}(h_w) + \frac{1}{\sqrt{n}}\left(KL(\rho||\pi) + \log\left(\frac{1}{\delta}\right) + \frac{(b-a)^2}{2}\right).$$

**Improved Lemma:** This bound can be improved slightly, since we can derive it directly from Theorem A.3. In particular for any fixed $w$ the random variable $l(h_w(X), Y)$ is bounded in $[a, b]$ and we know that $R_{RM}(h_w) = \mathbb{E}_{(X,Y) \sim \mathcal{D}} l(h_w(X), Y)$. Then, from Hoeffding's lemma:

$$\mathbb{E}_{(X,Y) \sim \mathcal{D}} \exp \lambda\left(R_{RM}(h_w) - l(h_w(X), Y)\right) \leq \exp\left(\frac{\lambda^2(b-a)^2}{8}\right), \forall \lambda \in \mathbb{R}.$$

Using the above result we find for every $w \in W$ and $\lambda \in \mathbb{R}$:

$$\mathbb{E}_{S \sim \mathcal{D}^n} \exp \lambda\left(R_{RM}(h_w) - R_{RM,n}(h_w)\right) = \mathbb{E}_{S \sim \mathcal{D}^n} \exp \frac{\lambda}{n}\left(\sum_{i=1}^n (R_{RM}(h_w) - l(h_w(X_i), Y_i))\right)$$

$$= \mathbb{E}_{S \sim \mathcal{D}^n} \prod_{i=1}^n \exp\left(\frac{\lambda}{n}(R_{RM}(h_w) - l(h_w(X_i), Y_i))\right) \overset{i.i.d}{=} \prod_{i=1}^n \mathbb{E}_{(X_i, Y_i) \sim \mathcal{D}} \exp\left(\frac{\lambda}{n}(R_{RM}(h_w) - l(h_w(X_i), Y_i))\right)$$

$$\leq \prod_{i=1}^n \exp\left(\frac{\lambda^2(b-a)^2}{8n^2}\right) = \exp\left(\frac{\lambda^2(b-a)^2}{8n}\right).$$

So, the Hoeffding assumption is satisfied with $f(\lambda, n) = \frac{\lambda^2(b-a)^2}{8n}$ and the PAC-Bayes bound becomes:

$$\mathbb{E}_{w \sim \rho} R_{RM}(h_w) \leq \mathbb{E}_{w \sim \rho} R_{RM,n}(h_w) + \frac{1}{\lambda}\left(KL(\rho||\pi) + \log\left(\frac{1}{\delta}\right) + \frac{\lambda^2}{n}\frac{(b-a)^2}{8}\right). \tag{11}$$

### A.2.2 Lipschitz Lemmas

We prove some key lemmas below. We denote $l(w; X, Y) = l(h_w(X), Y)$, $R_{RM}^l(w) = R_{RM}(h_w)$, and $R_{RM,n}^l(w) = R_{RM,n}(h_w)$.

**Lemma A.6.** *If the map $l(\cdot; X, Y)$ is $L$-Lipschitz continuous for any $(X, Y) \in \mathcal{X} \times \mathcal{Y}$, then both $R_{RM}^l(\cdot)$ and $R_{RM,n}^l(\cdot)$ are $L$-Lipschitz continuous.*

*Proof.* For any $(w, w') \in W^2$

$$|R_{RM}^l(w') - R_{RM}^l(w)| = |\mathbb{E}_{(X,Y)\sim\mathcal{D}}l(w'; X, Y) - \mathbb{E}_{(X,Y)\sim\mathcal{D}}l(w; X, Y)|$$
$$\leq \mathbb{E}_{(X,Y)\sim\mathcal{D}}|l(w'; X, Y) - l(w'; X, Y)| \leq \mathbb{E}_{(X,Y)\sim\mathcal{D}}L\|w' - w\| = L\|w' - w\|.$$

For $R_{RM,n}^l$, the analogous argument holds by replacing $\mathbb{E}_{(X,Y)\sim\mathcal{D}}$ with the empirical expectation $\frac{1}{n}\sum_{i=1}^n$. $\quad\square$

**Lemma A.7.** *If $l(\cdot; X, Y)$ is $L$-Lipschitz continuous for any $(X, Y) \in \mathcal{X} \times \mathcal{Y}$, then for any distribution $\rho$ on $W$ with $\mathbb{E}_{w\sim\rho}[w] = w_0 \in W$, it holds that:*

$$|R_{RM}^l(w_0) - \mathbb{E}_{w\sim\rho}R_{RM}^l(w)| \leq L\sqrt{\mathrm{trCov}_{w\sim\rho}[w]}.$$

*Similarly, for $R_{RM,n}^l$:*

$$|R_{RM,n}^l(w_0)) - \mathbb{E}_{w\sim\rho}R_{RM,n}^l(w)| \leq L\sqrt{\mathrm{trCov}_{w\sim\rho}[w]}.$$

*Proof.* Since $l(\cdot; X, Y)$ is $L$-Lipschitz for all $(X, Y) \in \mathcal{X} \times \mathcal{Y}$, from Lemma A.6 we know that $R_{RM}^l(\cdot)$ is $L$-Lipschitz. Thus, for $w_0 \in W$ and for any $w \in W$ we have:

$$|R_{RM}^l(w_0) - R_{RM}^l(w)| \leq L\|w_0 - w\| \implies$$
$$R_{RM}^l(w) - L\|w_0 - w\| \leq R_{RM}^l(w_0) \leq L\|w_0 - w\| + R_{RM}^l(w), \implies$$
$$\mathbb{E}_{w\sim\rho}R_{RM}^l(w) - L\mathbb{E}_{w\sim\rho}\|w_0 - w\| \leq R_{RM}^l(w_0) \leq \mathbb{E}_{w\sim\rho}R_{RM}^l(w) + L\mathbb{E}_{w\sim\rho}\|w_0 - w\| \implies$$
$$|R_{RM}^l(w_0) - \mathbb{E}_{w\sim\rho}R_{RM}^l(w)| \leq L\mathbb{E}_{w\sim\rho}\|w_0 - w\|.$$

From the Cauchy-Schwarz inequality, $\mathbb{E}_{w\sim\rho}\|w_0 - w\| \leq \sqrt{\mathbb{E}_{w\sim\rho}\|w_0 - w\|^2}$. Using that $\mathbb{E}_{w\sim\rho}[w] = w_0$ we have: $\mathbb{E}_{w\sim\rho}\|w - \mathbb{E}_{w\sim\rho}[w]\|^2 = \mathrm{trCov}_{w\sim\rho}[w]$. The desired result follows. The proof is similar for $R_{RM,n}$. $\quad\square$

Denote $R_{TRM}(w, \theta) = R_{TRM}(h_w, Q_\theta)$ and $R_{TRM,n}(w, \theta) = R_{TRM,n}(h_w, Q_\theta)$. Also define the *fixed-g loss* $l_g(w; X, Y) := l(h_w(gX), Y) = l(w; gX, Y)$ and the *fixed-g risk* $R_g(w) := R_{RM}(h_w \circ g)$ for all $g \in G_\Omega$.

**Lemma A.8.** *If $l(\cdot; X, Y)$ is $L$-Lipschitz continuous for any $(X, Y) \in \mathcal{X} \times \mathcal{Y}$ and a distribution $Q_w$ on $W$ with $\mathbb{E}_{w'\sim Q_w}[w'] = w \in W$, then it holds that:*

1. *$w \mapsto R_{TRM}(w, \theta)$ and $w \mapsto R_{TRM,n}(w, \theta)$ are $L$-Lipschitz continuous for all $\theta \in \Theta$.*

2. *$|R_{TRM}(w, \theta) - \mathbb{E}_{w'\sim Q_w}R_{TRM}(w', \theta)| \leq L\sqrt{\mathrm{trCov}_{w'\sim Q_w}[w']}$, $\forall(w, \theta) \in W \times \Theta$ and same for $R_{TRM,n}$.*

3. *In particular, if $Q_w = \mathcal{N}(w, \sigma^2 I)$ we find:*

$$|R_{TRM}(w, \theta) - \mathbb{E}_{w'\sim Q_w}R_{TRM}(w', \theta)| \leq L\sqrt{p}\sigma, \ \forall(w, \theta) \in W \times \Theta$$

   *and the analogous inequality also holds for $R_{TRM,n}$.*

*Proof.* First we observe that since $l(\cdot; X, Y)$ is $L$-Lipschitz for all $(X, Y) \in \mathcal{X} \times \mathcal{Y}$ and $gX \in \mathcal{X}$, we have that $l_g(\cdot)$ is $L$-Lipschitz for all $g \in G_\Omega$. Then, we observe that $R_g(w) = R_{RM}^{l_g}(w)$. By applying Lemma A.6 to $l_g(\cdot)$ we find that $R_g(\cdot)$ is $L$-Lipschitz for all $g \in G_\Omega$. Then, applying Lemma A.7 to $l_g$ and $Q_w$ we find:

$$|R_g(w) - \mathbb{E}_{w'\sim Q_w}R_g(w')| = |R_{RM}^{l_g}(w) - \mathbb{E}_{w'\sim Q_w}R_{RM}^{l_g}(w')| \leq L\sqrt{\mathrm{trCov}_{w'\sim Q_w}[w']}. \tag{12}$$

Next we observe that $R_{TRM}(w,\theta) = \mathbb{E}_{g\sim Q_\theta} R_g(w)$ thus for all $w, w' \in W$,

$$|R_{TRM}(w',\theta) - R_{TRM}(w,\theta)| = |\mathbb{E}_{g\sim Q_\theta} R_g(w') - \mathbb{E}_{g\sim Q_\theta} R_g(w)| =$$
$$|\mathbb{E}_{g\sim Q_\theta}(R_g(w') - R_g(w))| \le \mathbb{E}_{g\sim Q_\theta} |R_g(w') - R_g(w)| \le \mathbb{E}_{g\sim Q_\theta} L\|w' - w\| = L\|w' - w\|.$$

This shows that $w \mapsto R_{TRM}(w,\theta)$ is $L$-Lipschitz, for all $\theta \in \Theta$, With a similar proof we can also show that $w \mapsto R_{TRM,n}(w,\theta)$ is $L$-Lipschitz continuous for all $\theta \in \Theta$.

Also, using Eq. 12 and $Q_w = \mathcal{N}(w, \sigma^2 I)$,

$$|R_{TRM}(w,\theta) - \mathbb{E}_{w'\sim Q_w} R_{TRM}(w',\theta)| = |\mathbb{E}_{g\sim Q_\theta} R_g(w) - \mathbb{E}_{w'\sim Q_w}\mathbb{E}_{g\sim Q_\theta} R_g(w')|$$

$$= |\mathbb{E}_{g\sim Q_\theta} R_g(w) - \mathbb{E}_{g\sim Q_\theta}\mathbb{E}_{w'\sim Q_w} R_g(w')| \le \mathbb{E}_{g\sim Q_\theta} |R_g(w) - \mathbb{E}_{w'\sim Q_w} R_g(w')| \le L\sqrt{\mathrm{trCov}_{w'\sim Q_w}[w']} = L\sqrt{p}\sigma.$$

The argument for $R_{TRM,n}$ is similar. $\qquad\square$

For some set $A$, consider a parametrization such that each $a \in A$ corresponds to a transformation $g_a \in G_\Omega$ and each pair $(w,a) \in W \times A$ indexes a predictor $x \to (h_w \circ g_a)(x)$. We define the *fixed-parametrized-g risk* as

$$R_p(w,a) := R_{g_a}(w) = \mathbb{E}_{(X,Y)\sim\mathcal{D}}[l((h_w \circ g_a)(X), Y)] = R_{RM}(h_w \circ g_a).$$

Similarly, $R_{p,n}(w,a) := R_{RM,n}(h_w \circ g_a)$. Thus, instead of transformations acting on $X$ via $g_a X$, we view them as an extension of a hypothesis space containing models of the form $h_w \circ g_a$. We can now use the bound in Theorem A.3 on the space $W \times A$ for $R_p(w,a)$.

### A.2.3  PAC-Bayes Bound

*Proof of Theorem 3.1.* The proof continues with the following steps: First, by applying the PAC-Bayes bound on $R_p(w,a)$ on product posterior measures over $W \times A$, the TRM risk appears naturally, but in the form of an expectation over the models. We then use Lipschitz continuity and the Lemma A.8 that we proved to derive the desired bound on TRM. Since the bound holds simultaneously for all posteriors over $W \times A$, we further constrain it appropriately. We conclude the proof by computing the KL divergence terms for the transformation and the models. At the end of the argument, we discuss optimization over the free parameters.

**Step 1 (Bound on the expected TRM loss):** Consider the prior $P_p := P_\mathcal{H} \times P_A$ with $P_A = \prod_{i=1}^K P_{(i)}$ from Theorem 3.1, $P_\mathcal{H} = \mathcal{N}(w_0, s^2 I)$ for some fixed $w_0 \in W$, and posteriors $Q_{w,\theta} := Q_{\mathcal{H},w} \times Q_\theta$ on $W \times A$ with $Q_\theta := \prod_{i=1}^K Q_{(i)}$ as specified by the theorem, with $Q_{\mathcal{H},w} = \mathcal{N}(w, \sigma^2 I)$. Since $Q_{\mathcal{H},w} \ll P_\mathcal{H}$, it only remains to prove that $Q_\theta \ll P_A$, $\forall \theta \in \Theta$, which we do later in Step 3. Assuming the Hoeffding assumption holds with $f(\lambda, n)$ (bounded losses being an example), we apply the bound from Theorem A.3 to $\delta \in (0,1], \lambda > 0$ and $P_p$ to $R_p(w,a)$. We find that with probability at least $1-\delta$ over the sampling of $S = \{(X_i, Y_i)\}_{i=1}^n \sim \mathcal{D}^n$ we have simultaneously over any probability measure $Q_{w,\theta}$ that:

$$\mathbb{E}_{w'\sim Q_w}\mathbb{E}_{a\sim Q_\theta} R_p(w',a) \le \mathbb{E}_{w'\sim Q_w}\mathbb{E}_{a\sim Q_\theta} R_{p,n}(w',a) + \frac{1}{\lambda}\left(KL(Q_{w,\theta}||P_p) + \log\left(\frac{1}{\delta}\right) + f(\lambda,n)\right) \iff$$

$$\mathbb{E}_{w'\sim Q_w} R_{TRM}(w',\theta) \le \mathbb{E}_{w'\sim Q_w} R_{TRM,n}(w',\theta) + \frac{1}{\lambda}\left(KL(Q_{w,\theta}||P_p) + \log\left(\frac{1}{\delta}\right) + f(\lambda,n)\right). \qquad (13)$$

**Step 2 (Bound on TRM for Lipschitz losses):** Now using that $l(\cdot; X, Y)$ is $L$-Lipschitz for any $(X,Y) \in \mathcal{X} \times \mathcal{Y}$, as well as Lemma A.8, we can derive a bound on TRM:

$$R_{TRM}(w,\theta) \overset{Lem.A.8}{\le} \mathbb{E}_{w'\sim Q_w} R_{TRM}(w',\theta) + L\sqrt{\mathrm{trCov}_{w'\sim Q_w}[w']}$$

$$\overset{Eq.(13)}{\le} \mathbb{E}_{w'\sim Q_w} R_{TRM,n}(w',\theta) + \frac{1}{\lambda}\left(KL(Q_{w,\theta}||P_p) + \log\left(\frac{1}{\delta}\right) + f(\lambda,n)\right) + L\sqrt{\mathrm{trCov}_{w'\sim Q_w}[w']}$$

$$\overset{Lem.A.8}{\le} R_{TRM,n}(w,\theta) + \frac{1}{\lambda}\left(KL(Q_{w,\theta}||P_p) + \log\left(\frac{1}{\delta}\right) + f(\lambda,n)\right) + 2L\sqrt{\mathrm{trCov}_{w'\sim Q_w}[w']}.$$

Now, considering $Q_w = \mathcal{N}(w, \sigma^2 I)$, we have:

$$R_{TRM}(w, \theta) \leq R_{TRM,n}(w, \theta) + \frac{1}{\lambda}\left(KL(Q_{w,\theta}||P_p) + \log\left(\frac{1}{\delta}\right) + f(\lambda, n)\right) + 2L\sqrt{p}\sigma.$$

Since the bound holds for any $\sigma > 0$, we can optimize over this parameter. Moreover, as the bound holds simultaneously for all $Q_{w,\sigma} = \mathcal{N}(w, \sigma^2 I)$, when we optimize the bound, $\sigma$ is permitted to depend on the optimization parameters. We will discuss optimization over $\sigma, \lambda$ after finding the form of the KL-divergence terms.

**Step 3 (KL divergence terms):** Now we will compute the $KL(Q_{w,\theta}||P_p)$ term. We have

$$KL(Q_{w,\theta}||P_p) = KL(Q_{\mathcal{H},w} \times \prod_{i=1}^{K} Q_{(i)}||P_{\mathcal{H}} \times \prod_{i=1}^{K} P_{(i)}) = KL(Q_{\mathcal{H},w}||P_{\mathcal{H}}) + \sum_{i=1}^{K} KL(Q_{(i)}||P_{(i)}).$$

We start with $KL(Q_{\mathcal{H},w}||P_{\mathcal{H}})$. Since $Q_{\mathcal{H},w} = \mathcal{N}(w, \sigma^2 I)$ and $P_{\mathcal{H}} = \mathcal{N}(w_0, s^2 I)$ it is known that

$$KL(Q_{\mathcal{H},w}||P_{\mathcal{H}}) = \frac{1}{2}\left[2p\log\frac{s}{\sigma} - p + \frac{1}{s^2}\|w - w_0\|^2 + \frac{\sigma^2}{s^2}p\right].$$

Then, we need to derive the expressions for each $KL(Q_{(i)}||P_{(i)})$. If $i \leq k$, then the base distributions are parametric, and we denote the KL term by $K_i(\alpha_i, \pi_i)$; while for $k < i \leq K$ the base distributions are parameter-free, and we denote the KL term $K_i(\pi_i)$.

We start with the more challenging $K_i(\alpha_i, \pi_i)$ term. We will omit the index $i$ to keep the notation uncluttered. Since none of the $Q_{(i)}, P_{(i)}$ are absolutely continuous with respect to the Lebesgue measure, we need to resort to the general definition of KL-divergence using the Radon-Nikodym derivative.

Let $(\mathbb{R}, \mathcal{L})$ be our measurable space. We will use the following definitions in the proof:

1. Dirac measure $\delta_0$, where $\delta_0(E) = \mathbb{1}[0 \in E]$ for all measurable sets $E$.
2. Lebesgue measure $\lambda$ on the real line.
3. Uniform measure $\mathrm{U}[-A, A]$, where $\mathrm{U}[-A, A](E) = \frac{\lambda(E \cap [-A, A])}{2A}$, for all measurable sets $E$.
4. Prior measure $P = \beta\delta_0 + (1 - \beta)\mathrm{U}[-A, A]$, where $A > 0$, $\beta \in (0, 1)$.
5. Posterior measure $Q = (1 - \pi)\delta_0 + \pi\mathrm{U}[-\alpha, \alpha]$, where $\alpha \in (0, A]$.

Both $P$ and $Q$ are $\sigma$-finite measures, since they are probability measures. Recall that a measure $\mu$ is absolutely continuous with respect to another measure $\nu$, written as $\mu \ll \nu$, if for all measurable sets $A$ such that $\nu(A) = 0$, we also have $\mu(A) = 0$. Here $\delta_0, \mathrm{U}[-A, A]$, and $Q$ are absolutely continuous with respect to $P$. Indeed, if there is some measurable set $E$ such that $P(E) = 0$, then

$$\beta\delta_0(E) + (1 - \beta)\mathrm{U}[-A, A](E) = 0 \implies \delta_0(E) = \mathrm{U}[-A, A](E) = 0$$

since $\beta \in (0, 1)$. Moreover, $Q(E) = (1 - \pi)\delta_0(E) + \pi\mathrm{U}[-\alpha, \alpha](E) = 0$, where we also used that $[-\alpha, \alpha] \subseteq [-A, A]$ which implies $\mathrm{U}[-\alpha, \alpha] \ll \mathrm{U}[-A, A]$.

However, $P$ and $Q$ are not absolutely continuous with respect to the Lebesgue measure, since $Q(\{0\}), P(\{0\}) > 0$ while $\lambda(\{0\}) = 0$. However, $\delta_0$ and $\mathrm{U}[-\alpha, \alpha]$ are absolutely continuous with respect to $P$, and thus we have:

$$\frac{dQ}{dP} = \frac{d((1 - \pi)\delta_0 + \pi\mathrm{U}[-\alpha, \alpha])}{dP} = (1 - \pi)\frac{d\delta_0}{dP} + \pi\frac{d\mathrm{U}[-\alpha, \alpha]}{dP}.$$

We can readily verify that the Radon-Nikodym derivatives have the form

$$\frac{d\delta_0}{dP}(x) = \frac{1}{\beta}\mathbb{1}[x = 0], \qquad \frac{d\mathrm{U}[-\alpha, \alpha]}{dP}(x) = \frac{A}{\alpha}\mathbb{1}[x \in [-\alpha, \alpha] \setminus \{0\}].$$

where the equalities hold $P$-almost everywhere. Indeed, we can check that these functions satisfy the required conditions, and then use the uniqueness of the Radon-Nikodym derivative up to $P$-null sets. Next we compute the Kullback-Leibler divergence $KL(Q||P)$.

$$
\begin{aligned}
KL(Q||P) &= \int_{\mathbb{R}} \log \frac{dQ}{dP} dQ = \int_{\mathbb{R}} \log \frac{dQ}{dP} d[(1-\pi)\delta_0 + \pi \mathrm{U}[-\alpha, \alpha]] \\
&= (1-\pi) \int_{\mathbb{R}} \log \frac{dQ}{dP} d\delta_0 + \pi \int_{\mathbb{R}} \log \frac{dQ}{dP} d\mathrm{U}[-\alpha, \alpha] \\
&= (1-\pi) \log \frac{dQ}{dP}(0) + 2\pi \int_{\mathbb{R}} \mathbb{1}[x \in [-\alpha, 0)] \log \frac{dQ}{dP} d\mathrm{U}[-\alpha, \alpha].
\end{aligned}
$$

Now, using our formulas for $\frac{dQ}{dP}$, $\frac{d\delta_0}{dP}$, and $\frac{d\mathrm{U}[-\alpha,\alpha]}{dP}$, we find

$$
\begin{aligned}
(1-\pi) \log \frac{1-\pi}{\beta} + 2\pi \frac{1}{2} \log \frac{\pi A}{(1-\beta)\alpha} &= (1-\pi) \log \frac{1-\pi}{\beta} + \pi \log \frac{\pi}{1-\beta} + \pi \log \frac{A}{\alpha} \\
&= KL(\mathcal{B}(\pi)||\mathcal{B}(1-\beta)) + \pi KL(\mathrm{U}[-\alpha, \alpha]||\mathrm{U}[-A, A]),
\end{aligned}
$$

where $\mathcal{B}(\pi)$ denotes the Bernoulli distribution with parameter $\pi$.

For $K_i(\pi_i)$, omitting again the index $i$ to keep the notation clean, we find

$$
\begin{aligned}
Q &= (1-\pi)\delta_0 + \pi \mathrm{U}[\{0, \ldots, N-1\}] = \left\{ 1 - \pi + \frac{\pi}{N}, \frac{\pi}{N}, \ldots, \frac{\pi}{N} \right\} \\
P &= \beta\delta_0 + (1-\beta)\mathrm{U}[\{0, \ldots, N-1\}] = \left\{ \beta + \frac{1-\beta}{N}, \frac{1-\beta}{N}, \ldots, \frac{1-\beta}{N} \right\}.
\end{aligned}
$$

Then,

$$
\begin{aligned}
KL(Q||P) &= \sum_{x=0}^{N-1} q(x) \log \frac{q(x)}{p(x)} = (1 - \pi + \frac{\pi}{N}) \log \frac{(1 - \pi + \frac{\pi}{N})}{(\beta + \frac{1-\beta}{N})} + (N-1)\frac{\pi}{N} \log \frac{\pi/N}{(1-\beta)/N} \\
&= (1 - \frac{(N-1)\pi}{N}) \log \frac{(1 - \frac{(N-1)\pi}{N})}{(1 - \frac{(N-1)(1-\beta)}{N})} + \frac{(N-1)\pi}{N} \log \frac{(N-1)\pi/N}{(N-1)(1-\beta)/N} \\
&= KL \left( \mathcal{B} \left( \left( 1 - \frac{1}{N} \right) \pi \right) || \mathcal{B} \left( \left( 1 - \frac{1}{N} \right)(1 - \beta) \right) \right).
\end{aligned}
$$

**Step 4 (Putting it all together):** We write the whole bound for completeness. Given $\delta \in [0, 1), \lambda > 0$ and a prior measure $P_{\mathcal{H}} \times \prod_{i=1}^{K} P_{(i)}$ it holds with probability at least $1 - \delta$ over the draw of $S = \{(X_i, Y_i)\}_{i=1}^{n} \sim \mathcal{D}$ and simultaneously over $w \in W$ and $\theta \in \Theta$, where $\theta = \{a_1, \ldots, a_k, \pi_1, \ldots, \pi_K\}$ and $a_i \in (0, A_i]$ and $\pi_i \in [0, 1]$, that:

$$
R_{TRM}(w, \theta) \leq R_{TRM,n}(w, \theta) +
$$
$$
\frac{1}{\lambda} \left( \frac{1}{2} \left[ 2p \log \frac{s}{\sigma} - p + \frac{1}{s^2} \|w - w_0\|^2 + \frac{\sigma^2}{s^2} p \right] + \mathrm{Reg}(\theta) + \log \left( \frac{1}{\delta} \right) + f(\lambda, n) \right) + 2L\sqrt{p}\sigma, \quad (14)
$$

where $\mathrm{Reg}(\theta)$ is defined in equation 5 in Theorem 3.1.

**TRM bound for a countable $W$:** In the above analysis, Lipschitz continuity is used to move from a probabilistic version of TRM to a deterministic one. However, Lipschitz continuity of the loss is not needed is when $W$ is countable. In that case, by choosing $Q_{\mathcal{H},w} = \delta_w$ where $\delta_w$ is the discrete Dirac mass on $h_w$, the KL divergence term is equal to: $\log \left( \frac{1}{P_{\mathcal{H}}(w)} \right) + \sum_{i=1}^{K} KL(Q_{(i)}||P_{(i)})$, and the bound is:

$$
R_{TRM}(w, \theta) \leq R_{TRM,n}(w, \theta) + \frac{1}{\lambda} \left( \log \left( \frac{1}{P_{\mathcal{H},w}\delta} \right) + \sum_{i=1}^{K} KL(Q_{(i)}||P_{(i)}) + f(\lambda, n) \right). \quad (15)
$$

**Optimization over the "hyperparameters" $\sigma, \lambda$:**

1. **Optimization over** $\sigma > 0$: As we discussed above, since the bound holds for any $\sigma > 0$, we can optimize over this variable before optimizing with respect to $w$. From the right hand side of Eq. 14, we see that the terms involving $\sigma$ are:

$$-\frac{p}{\lambda}\log\sigma + \frac{p}{2\lambda s^2}\sigma^2 + 2L\sqrt{p}\sigma.$$

This is a convex function of $\sigma$, attaining its minimum at $\sigma^*(\lambda) = \left(\sqrt{\frac{\lambda^2 L^2}{p} + \frac{1}{s^2}} + \frac{\lambda L}{\sqrt{p}}\right)^{-1}$.

The bound would hold simultaneously for all $\sigma > 0$ if we considered posteriors of the form $Q_{\mathcal{H},w,\sigma} = \mathcal{N}(w, \sigma^2 I)$. Then, $\sigma$—which also appears in the "training error" term—would be permitted to depend on the parameters during optimization.

2. **Optimization with respect to** $\lambda$: The bounds we derived hold for any $\lambda > 0$. However, they do not hold uniformly over $\lambda$. In practice, this implies that we have to set some fixed $\lambda$ in the beginning before optimizing over the parameters of the posterior. In particular, it is not permitted for $\lambda$ to depend on these parameters. In some cases, we can optimize the right hand side of the bound over $\lambda$, set $\lambda$ to this value, and then optimize over the parameters. This is the case for example in Eq. 11 when $KL(\rho||\pi)$ does not depend on $\rho$. For example if $\rho(w) = \delta_w$ on a countable space $W$, then $KL(\rho||\pi) = \log\frac{1}{\pi(w)}$ in which case $\lambda^* = \frac{1}{b-a}\sqrt{8n\log\frac{1}{\pi(w)\delta}}$. In our case, even for this simple scenario discussed in equation 15, optimization over $\lambda$ is still not possible, since the KL-divergence also depends on the parameters $\theta$ of our distribution of transformations $Q_\theta$.

There are various techniques to address the dependence on $\lambda$. Typically, a union bound over a discrete grid $\Lambda$ is used to obtain the PAC-Bayes bound uniformly over $\lambda$ on this grid. This loosens the initial bound by a logarithmic term $\frac{\log|\Lambda|}{n}$ in the cardinality of the grid. Langford & Caruana (2002) choose a grid to loosen the bound only by an iterated logarithm factor $\frac{\log\log|\Lambda|}{n}$ in the cardinality of the grid. Moreover, they relax the upper bound even further to transform the discrete grid into a continuous grid. Optimizing over a discrete grid requires multiple training loops, which is not efficiently feasible in our application. Moreover, gradient updates jointly on the parameters of the model, the transformations, and the regularization parameter $\lambda$ may result in unstable training, which moreover can also depend strongly on the initialization.

Another method is to use a bound that is not dependent on any parameter $\lambda$, such as the bound in (McAllester, 1999). However, that has been derived only for bounded losses. That bound suggests an objective similar to the one we are currently using.

Experimentally, we found out that $\lambda = \sqrt{n}$ works well in practice. Then, using the optimal $\sigma^*(\lambda) \approx \frac{1}{\sqrt{n}}\frac{\sqrt{p}}{2L}$ the bound becomes (e.g., for subgaussian losses with variance factor $s^2$):

$$R_{TRM}(w,\theta) \leq R_{TRM,n}(w,\theta) +$$
$$\frac{1}{\sqrt{n}}\left(\frac{1}{2}\left[2p\log\frac{s}{\sigma} - p + \frac{1}{s^2}\|w - w_0\|^2 + \frac{\sigma^2}{s^2}p\right] + \text{Reg}(\theta) + \log\left(\frac{1}{\delta}\right) + \frac{s^2}{2}\right) + 2L\sqrt{p}\sigma$$
$$= R_{TRM,n}(w,\theta) + \frac{1}{\sqrt{n}}\left(\frac{1}{2s^2}\|w - w_0\|^2 + \text{Reg}(\theta)\right) + c_n \tag{16}$$

where $c_n = \frac{1}{\sqrt{n}}\left(p\log\frac{2Ls\sqrt{n}}{\sqrt{p}} + \frac{1}{n}\frac{p^2}{8L^2s^2} + \log\frac{1}{\delta} + \frac{s^2}{2} + \frac{p}{2}\right) \xrightarrow{n\to\infty} 0$.

$\square$

**Remark A.9.** *As mentioned in Dziugaite & Roy (2017), bounding the risk $R_{RM}(w)$ with respect to the parameters $w \in W$ instead of the elements $h_w \in \mathcal{H}$ might result in looser bounds due to overparametrization. Much tighter bounds would be obtained if we considered $\mathcal{H}$ as the quotient space $W/\sim$, where the equivalence relation $\sim$ is defined as $w \sim w' \iff h_w = h_{w'}$ and re-derived the bound directly on this quotient space for $R_{RM}(h_w)$. However, finding these equivalences in a deep network may in general be intractable. In our setup,*

*we would further tighten the bounds if we derived them for $Q$ supported on transformations $\rho \in G_Q$—namely $R_{TRM}(h_w, Q_\theta)$—rather than on the parameters of the transformations $a \in A$, namely $R_{TRM}(w, a)$.*

*We could do that in two steps. First, for each base distribution $Q_{(i)}$ on $A_{(i)}$, consider $G_{Q_{(i)}}$ as the space $A_{(i)}/\sim$ where $a \sim a' \iff \rho(a) = \rho(a')$. Since the base distributions are usually simple, we could actually establish an isomorphism in this case by careful construction of $A_{(i)}$ (e.g., rotations not exceeding $2\pi$). As a second step, consider the distributions $Q$ on the space $A$, and view $G_Q$ as $A/\sim$, where for $\rho(a) := \rho_1(a_1) \circ \ldots \rho_K(a_K) \in G_Q$ we have $a \sim a' \iff \rho(a) = \rho(a')$. There are practical scenarios where an isomorphism can be established in this case too, depending on the structure of the group. For example, when the assumptions on the base transformations above hold, and the total transformation space is a inner semi-direct product of a larger group, i.e., $G_Q = G_{Q_{(1)}} \ldots G_{Q_{(K)}}$, the space of compositions can be identified with the Cartesian product.*

*Further, we could also consider all posteriors $Q \in \Omega$, instead of the posteriors indexed by $\Theta$, i.e., $\theta \sim \theta' \iff Q_\theta = Q'_\theta$. In our example with mixtures of base distributions, $\Omega$ and $\Theta$ are actually isomorphic given the conditions above and if not all parameter-free bases are point masses at identity. Finally, one more improvement could be to derive the bounds for posteriors on $\mathcal{H} \circ G_Q$, namely $R_{RM}(h_w \circ g_a)$ instead of the product space $\mathcal{H} \times G_Q$, i.e., the map $(h_w, g_a) \to R_{RM}(h_w \circ g_a)$.*

## A.3   Unbiased Gradient Estimates

In Eq. 6 we define the loss for SCALE as:

$$L(w, \theta) := \frac{1}{n} \sum_{i=1}^{n} \mathbb{E}_{g \sim Q_\theta} \left[ l(h_w(gx_i), y_i) \right] + \lambda_{\text{reg}} \text{Reg}(\theta)$$

with $\theta = \{\pi_1, \ldots, \pi_K, \alpha_1, \ldots, \alpha_k\}$. We note again that, as stated in 3.2, from the $K$ base transformation distributions we assume that only the first $k$ contain learnable parameters $\alpha_i$, $i \in [k]$, respectively.

Computing the gradient of the regularization term $\lambda_{\text{reg}} \text{Reg}(\theta)$ is straightforward, so we will focus on the gradient computation for the first term of the loss. For a fixed $s \in \{1, \ldots, K\}$, we can write the first term of the loss as:

$$\frac{1}{n} \sum_{i=1}^{n} \mathbb{E}_{g_1 \sim Q_{(1)}} \ldots \mathbb{E}_{g_s \sim Q_{(s)}} \ldots \mathbb{E}_{g_K \sim Q_{(K)}} [l(h_w(g_1 \ldots g_s \ldots g_K x_i), y_i)] =$$

$$\frac{1}{n} \sum_{i=1}^{n} \left( (1 - \pi_s) \cdot \mathbb{E}_{g_1 \sim Q_{(1)}} \ldots \mathbb{E}_{g_K \sim Q_{(K)}} \left[ l(h_w(g_1 \ldots g_{s-1} g_{s+1} \ldots g_K x_i), y_i) \right] \right.$$

$$\left. + \pi_s \cdot \mathbb{E}_{g_1 \sim Q_{(1)}} \ldots \mathbb{E}_{g_s \sim U_{s,\alpha_s}} \ldots \mathbb{E}_{g_K \sim Q_{(K)}} \left[ l(h_w(g_1 \ldots g_s \ldots g_K x_i), y_i) \right] \right). \tag{17}$$

The partial derivative with respect to $\pi_s$ is

$$\frac{1}{n} \sum_{i=1}^{n} \left( - \mathbb{E}_{g_1 \sim Q_{(1)}} \ldots \mathbb{E}_{g_K \sim Q_{(K)}} \left[ l(h_w(g_1 \ldots g_{s-1} g_{s+1} \ldots g_K x_i), y_i) \right] \right.$$

$$\left. + \mathbb{E}_{g_1 \sim Q_{(1)}} \ldots \mathbb{E}_{g_s \sim U_{s,\alpha_s}} \ldots \mathbb{E}_{g_K \sim Q_{(K)}} \left[ l(h_w(g_1 \ldots g_s \ldots g_K x_i), y_i) \right] \right). \tag{18}$$

To compute an unbiased estimator of this gradient we first sample $M$ iid copies from the distribution of $g = g_1 \ldots g_{s-1} u_s g_{s+1} \ldots g_K$, where each $g_\chi$ with $\chi \neq s$ is sampled from $Q_{(\chi)}$, and $u_s$ is sampled from $U_{s,\alpha_s}$. We denote by $g_\chi^{(j)}$ and $u_s^{(j)}$ the $j^{th}$ sample from the distribution of $g_\chi$ and $u_s$, respectively. This leads to the following unbiased Monte Carlo estimator of the partial derivative with respect to $\pi_s$ shown in equation 18:

$$\frac{1}{nM} \sum_{i=1}^{n} \left( \sum_{j=1}^{M} l\left( h_w(g_1^{(j)} \ldots u_s^{(j)} \ldots g_K^{(j)} x_i), y_i \right) - \sum_{j=1}^{M} l\left( h_w(g_1^{(j)} \ldots g_{s-1}^{(j)} g_{s+1}^{(j)} \ldots g_K^{(j)} x_i), y_i \right) \right). \tag{19}$$

Now suppose that $s \in \{1, \ldots, k\}$ so the $s$-th transformation corresponds to a parametric distribution $Q_{(s)}$ with learnable parameter $a_s$. For the partial derivative with respect to $\alpha_s$, we assume that the sample $u_s \sim U_{s,\alpha_s}$ can be written as $u_s = \rho_s(a)$ with $a = \alpha_s \varepsilon, \varepsilon \sim \mathrm{U}[-1,1]$ and with $\rho_s$ being a differentiable transformation. We assume that all $g_\chi$ with $\chi < s$ are differentiable with respect to their input, i.e., as functions from $\mathcal{X}$ to $\mathcal{X}$. Importantly, this covers "discrete" transformations such as flips, which are differentiable transforms of the input object.

Under these assumptions, using the reparametrization trick (Kingma & Welling, 2014), which is essentially the same idea as functional/structural inference, see e.g., Fraser (1966), we find the partial derivative with respect to $\alpha_s$:

$$\frac{\partial}{\partial \alpha_s} \left( \frac{1}{n} \sum_{i=1}^{n} \mathbb{E}_{g_1 \sim Q_{(1)}} \ldots \mathbb{E}_{g_s \sim Q_{(s)}} \ldots \mathbb{E}_{g_K \sim Q_{(K)}} [l(h_w(g_1 \ldots g_s \ldots g_K x_i), y_i)] \right) =$$

$$\frac{\pi_s}{n} \sum_{i=1}^{n} \mathbb{E}_{g_1 \sim Q_{(1)}} \ldots \mathbb{E}_{\varepsilon \sim [-1,1]} \ldots \mathbb{E}_{g_K \sim Q_{(K)}} \left[ \frac{\partial l(h_w(g_1 \ldots \rho_s(\alpha_s \varepsilon) \ldots g_K x_i), y_i)}{\partial a_s} \right]. \tag{20}$$

Similarly to before, an unbiased estimator of the gradient from equation 20 can be computed by taking $M$ samples from the distribution of each $g_\chi \sim Q_{(\chi)}$ with $\chi \neq s$ and $M$ samples from the distribution $\varepsilon \sim \mathrm{U}[-1,1]$. This leads to the following unbiased Monte Carlo estimator of the partial derivative with respect to $\alpha_s$:

$$\frac{\pi_s}{nM} \sum_{i=1}^{n} \sum_{j=1}^{M} \left[ \frac{\partial l(h_w(g_1^{(j)} \ldots \rho_s(\alpha_s \varepsilon^{(j)}) \ldots g_K^{(j)} x_i), y_i)}{\partial \alpha_s} \right]. \tag{21}$$

**Efficient gradient estimation:** During training, for each datapoint $x$ we take $M$ samples from the distribution of $gx = \prod_{i=1}^{K} g_i x$. For each $g_\chi$, we first sample $t_\chi \sim \mathrm{Bern}(\pi_\chi)$ and $u_\chi \sim U_{\chi,\alpha_\chi}$ if $\chi \leq k$ or $u_\chi \sim U_\chi$ if $\chi > k$. Then we set $g_\chi = \mathbb{1}(t_\chi = 0)I + \mathbb{1}(t_\chi = 1)u_\chi$. This means that among the $M$ samples from the distribution of $gx$, the expected number of transforms $g_\chi$ sampled from the transformation distribution $U_{\chi,\alpha_\chi}$ or $U_\chi$ is $\pi_\chi M$. To obtain an efficient training procedure, we develop estimators for the derivatives with respect to *all* parameters of $Q_\theta$ using *only these $M$ samples*.

For $s \in \{1, \ldots, K\}$, we modify the estimator from equation 19 for the partial derivative with respect to $\pi_s$ (when $\pi_s \in (0,1)$) as follows:

$$\frac{1}{n} \sum_{i=1}^{n} \left( \frac{\sum_{j=1}^{M} \mathbb{1}(t_s^{(j)} = 1) \, l\left(h_w(g_1^{(j)} \ldots u_s^{(j)} \ldots g_K^{(j)} x_i), y_i\right)}{\sum_{j=1}^{M} \mathbb{1}(t_s^{(j)} = 1)} \right.$$

$$\left. - \frac{\sum_{j=1}^{M} \mathbb{1}(t_s^{(j)} = 0) \, l\left(h_w(g_1^{(j)} \ldots g_{s-1}^{(j)} g_{s+1}^{(j)} \ldots g_K^{(j)} x_i), y_i\right)}{\sum_{j=1}^{M} \mathbb{1}(t_s^{(j)} = 0)} \right). \tag{22}$$

For $s \in \{1, \ldots, k\}$ we modify the partial derivative with respect to $\alpha_s$ as follows:

$$\frac{1}{nM} \sum_{i=1}^{n} \sum_{j=1}^{M} \mathbb{1}(t_s^{(j)} = 1) \left[ \frac{\partial l(h_w(g_1^{(j)} \ldots \rho_s(\alpha_s \varepsilon^{(j)}) \ldots g_K^{(j)} x_i), y_i)}{\partial \alpha_s} \right]. \tag{23}$$

One can readily verify that these estimators remain consistent as the number $M$ of samples tends to infinity.

The estimator from equation equation 22 can be problematic when $\pi_s$ is very close to zero or unity, which results in $\mathbb{1}(t_s = 1)$ or $\mathbb{1}(t_s = 0)$ being zero with high probability. When this happens, we set the corresponding term in the estimator (i.e., the fraction whose denominator is zero), to zero. In addition, to avoid this scenario, especially early on in the training process, before convergence, we—somewhat heuristically—constrain $\pi_s$ to the interval $[c, 1-c]$ for some $c \in (0, 0.5)$, as we will describe in Section A.4. During training, we reduce $c$ linearly, so that at the end of training $c$ is effectively zero.

### A.4 Training Details using SCALE

When we train using SCALE, we optimize both the network parameters and the parameters of the distribution $Q_\theta$ in a single training process. We have generally observed that the algorithm is not too sensitive to the hyperparameters, and a relatively straightforward and quick tuning process (say, a small grid search over a few values) is enough to find good defaults. The experiments were executed using GeForce RTX 2080 Ti GPUs.

First we present the hyperparameters that are kept fixed in all datasets. We found these hyperparameters by a grid search to ensure a stable convergence behavior across multiple datasests. We initialize all $\alpha_i$ to 0.1 and all $\pi_i$ to $1/K$, where recall that $K$ is the number of transformations composed. Additionally, to avoid having $\pi_i$ converge to zero or unity at the beginning of the training, we constraint all $\pi_i$ to the interval $[c, 1-c]$. We initialize $c = 0.4/K$ and reduce it linearly (by a constant after each epoch) so that at the end of the training it equals zero. Since we have a pre-determined number of epochs for each dataset (discussed below), this determines the amount to decrease after each epoch. For the computation of the gradient presented in A.3, we observe that with $M \geq 4$, the training stays nearly unaffected by the number of Monte Carlo samples $M$. We use $M = 4$ to reduce the required computation. For MNIST/rotMNIST we use a batch size of 128, while for CIFAR 10/100 we use a batch size of 64.

Finally we use the following network-specific hyperparameters:

- **MNIST/rotMNIST:** We use the same network architecture used in Benton et al. (2020), which consists of five convolutional layers and one fully connected layer. To optimize the network's parameters we use the Adam optimizer (Kingma & Ba, 2015) with a learning rate of 0.02 and with cosine annealing Loshchilov & Hutter (2017). For the regularization term, in the rotMNIST and MNIST experiments $\lambda_{\text{reg}}$ is set to 0.006. To learn the augmentation parameters $\pi_i$, after each epoch, we linearly reduce the learning rate from 0.001 to zero in order to make the training more robust to the choice of the hyperparameter $\lambda_{\text{reg}}$. We train for 200 epochs by jointly optimizing the parameters of $Q_\theta$ and the parameters of the network, then we train for another 100 epochs by optimizing only the parameters of the network.

- **CIFAR 10/100:** We use a WideResNet 28-10 (Zagoruyko & Komodakis, 2016). We optimize the parameters of the network with an SGD optimizer with learning rate of 0.1, cosine annealing, weight decay of 0.0001 and Nesterov momentum (Nesterov, 1983) with value 0.9. For the CIFAR 10 experiment, $\lambda_{\text{reg}}$ is set to 0.01, for CIFAR 100 $\lambda_{\text{reg}}$ is set to 0.02. To learn the augmentation parameters $\pi_i$, after each epoch, we linearly reduce the learning rate from 0.001 to zero. We train for 200 epochs by jointly optimizing both the parameters $Q_\theta$ and the network parameters.

For all reported results (when no test-time augmentation is applied), we present the average test accuracy over 5 trials over all elements of randomness in the training (random initialization, random sampling of transforms, etc). The standard deviation of the test accuracy is 0.12% for the RotMNIST experiments, 0.09% for the CIFAR 10 experiments, and 0.1% for CIFAR 100.

Additionally, when we apply test-time augmentation, there is an additional variance in the model accuracy due to the randomly sampled augmentations during inference. Table 3 shows the mean accuracy and the standard deviation in the results due to the random choice of test-time augmentations. Again, we can see that the variance of the test accuracy is small enough that it does not affect the quantitative comparison between the models. Finally Figure 7 shows the final parameters $\pi_i$, $\alpha_i$ learned using SCALE on CIFAR10/100.

### A.5 Implementation Details for Parametrization

In the parametrization presented in Section 5.1 we include the following transformations: rotation (both a continuous version and a discrete rotation by $180^o$), $X$-axis scaling, $Y$-axis scaling, $X$-axis shearing, horizontal flip, and random cropping. We implement the continuous rotation, scaling, and shearing using the differentiable image sampling method proposed in Spatial Transformers Networks by Jaderberg et al. (2015). We implement each of the above transformations as follows:

Table 3: Test (Mean accuracy/ Standard deviation) of a Wide ResNet 28-10 trained on CIFAR 10/100, using various methods for learning data augmentations. All methods are evaluated using test-time augmentation with $n = 4, 8$ samples. The standard deviation is computed using five trials over the random choice of test-time augmentations.

| | | AUGERINO +FLIPS | FAST-AA | FASTER-AA | DADA GEOMETRIC | SCALE (Ours) |
|---|---|---|---|---|---|---|
| CIFAR10 | TESTAUG. (N=4) | 95.7%/ 0.1% | **97.0**%/ 0.08% | 96.6%/ 0.09% | 96.5%/0.1% | 96.7%/ 0.09% |
| | TESTAUG. (N=8) | 95.8%/ 0.1% | **97.2**%/ 0.04% | 96.8%/ 0.07% | 96.8%/0.1% | 96.9%/ 0.07% |
| CIFAR100 | TESTAUG. (N=4) | 79.2%/ 0.2% | 82.5 %/ 0.1% | 82.0%/ 0.09% | 79.8%/ 0.1% | **83.0**%/ 0.12% |
| | TESTAUG. (N=8) | 79.3%/ 0.2% | 82.9%/ 0.08% | 82.4%/ 0.1% | 80.1%/ 0.1% | **83.3**%/ 0.1% |

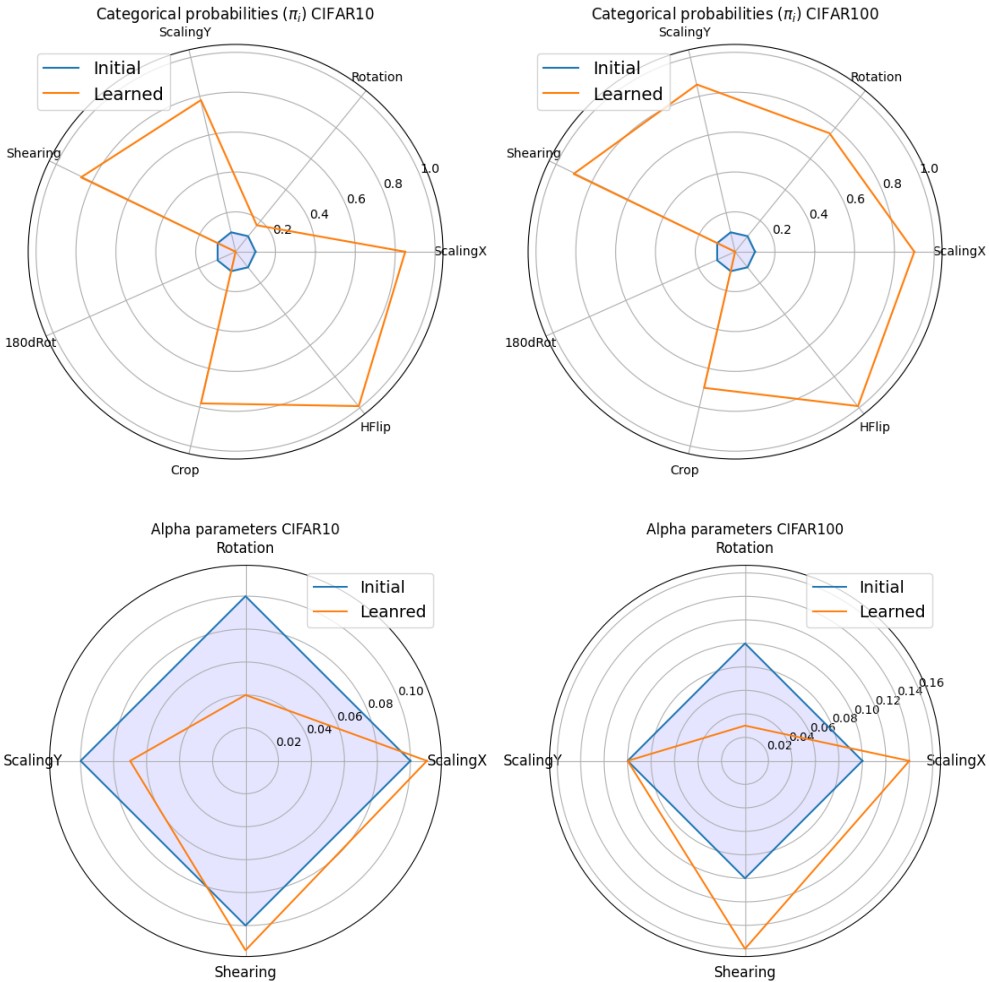

Figure 7: Initial values of $\pi_i$, $\alpha_i$ and the final parameters learned after training with SCALE on CIFAR10 and CIFAR100

- Rotation: For the continuous rotation, we sample the angle $a_1$ (in radians) by sampling $\varepsilon \sim \mathrm{U}[-1, 1]$ and taking $a_1 = \alpha_1 \varepsilon$. Then we perform the rotation using the following rotation matrix:

$$\begin{bmatrix} \cos(a_1) & -\sin(a_1) & 0 \\ \sin(a_1) & \cos(a_1) & 0 \\ 0 & 0 & 1 \end{bmatrix}. \tag{24}$$

- Discrete rotation by $180^o$: This is performed using the same rotation matrix from equation 24 as the general continuous rotation, with angle $a_5$ replacing $a_1$ in equation 24. The angle $a_5$ is sampled uniformly from $\{0, \pi\}$.

- $X$-axis scaling: Similar to the rotation, we sample $a_2 = \alpha_2\varepsilon$ by sampling $\varepsilon \sim \mathrm{U}[-1,1]$. Then the scaling is performed using the transformation matrix:

$$\begin{bmatrix} e^{a_2} & 0 & 0 \\ 0 & 1 & 0 \\ 0 & 0 & 1 \end{bmatrix}.$$

- $Y$-axis scaling: We sample $a_3 = \alpha_3\varepsilon$ by sampling $\varepsilon \sim \mathrm{U}[-1,1]$. Then the scaling is performed using the transformation matrix:

$$\begin{bmatrix} 1 & 0 & 0 \\ 0 & e^{a_3} & 0 \\ 0 & 0 & 1 \end{bmatrix}.$$

- X-axis shearing: We sample $a_4 = \alpha_4\varepsilon$ by sampling $\varepsilon \sim \mathrm{U}[-1,1]$. Then the shearing is performed using the transformation matrix:

$$\begin{bmatrix} 1 & a_4 & 0 \\ 0 & 1 & 0 \\ 0 & 0 & 1 \end{bmatrix}.$$

- Horizontal flip: To perform the horizontal flip, we sample $a_6$ uniformly from $\{-1, 1\}$. If $a_6 = -1$, then the order of the pixels on the $X$-axis is flipped, otherwise the input image stays unchanged. To combine the flips more efficiently with the rotations, scalings, and shearings, we use the transformation matrix

$$\begin{bmatrix} a_6 & 0 & 0 \\ 0 & 1 & 0 \\ 0 & 0 & 1 \end{bmatrix}.$$

- Random Crops: For the random cropping we use the implementation from the Torchvision package within PyTorch (Paszke et al., 2019) with padding equal to four.

## A.6 SVHN Experiment: Precomputed policies are not transferable

Precomputed augmentation policies that are produced by computationally expensive search methods similar to Fast-AA are available only for a limited set of the most frequently used datasets. Additionally, there is no guarantee that we can succesfully transfer these policies to new datasets. As a result, for any new dataset, a user needs to search for a new best policy, which highlights the importance of methods with low computational cost.

In this section we perform experiments on the SVHN dataset (Netzer et al., 2011), showcasing that precomputed policies *are not perfectly transferable even between similar datasets*. SVHN contains three sets of samples: the "train" set containing 73,257 difficult training samples, the "extra" set containing 531,131 less difficult training samples and the "test" set containing 26,032 test samples. For the precomputed policy, we use a Fast-AA policy computed for the whole SVHN dataset ("train" set and "extra" set), to train a network on a smaller training set that contains only the more difficult training samples ("train" set).

Table 4 compares the accuracies achieved using SCALE, Augerino, DADA (with the same augmentation space) and the precomputed Fast-AA policy to train on the "train" set of SVHN. We show that SCALE outperforms both the precomputed Fast-AA policy and Augerino and DADA.

For this experiment we used a ResNet34 network (He et al., 2016) trained with learning rate 0.001, cosine annealing and the Adam optimizer (Kingma & Ba, 2015). For the SCALE algorithm we used $\lambda_{\mathrm{reg}}$ initialized at 0.02 and for Augerino we used $\lambda_{\mathrm{reg}} = 0.05$.

Table 4: Test Accuracy on SVHN when the model is trained on the "train" set using Augerino, the Fast-AA policy (precomputed for the whole SVHN), DADA and SCALE. We show the average accuracy over 5 trials. For all methods the standard deviation is less than 0.12%

| | AUGERINO | FAST-AA (PRECOMPUTED POLICY) | DADA GEOMETRIC | SCALE (OURS) |
|---|---|---|---|---|
| SVHN | 97.2% | 97.1% | 96.6% | **97.4%** |

## A.7   A Bayesian Perspective of ETRM

In this section, we will see how variational inference (MacKay, 2003; Wainwright & Jordan, 2008) on a specific graphical model suggests optimizing a similar objective as the one we arrived to in Theorem 3.1 using PAC-Bayes bounds. By optimizing this objective we find the model $h_w \in \mathcal{H}$ and distribution $Q \in \Omega$ that minimize a tradeoff between the ETRM loss (equation 3) and negative entropy.

First we build the corresponding graphical model. To simplify the discussion we assume that all random variables have a density. A reasonable analogue in the setting of graphical models to the transformation hypothesis space $\Omega$ in TRM is a corresponding prior on the transformations $g : \mathcal{X} \to \mathcal{X}$. In particular, before seeing any data, we can assume $p(g)$ is the uniform density on $G_\Omega$; this choice can be viewed as a non-informative prior that maximizes the prior entropy. Instead of the standard observation model—or conditional density—$p(y|x)$ in supervised learning, our observation model now is $p(y|g, x)$. During training, we assume that the functional form of this conditional distribution is known, but depends on unknown parameters $w$. For example, in binary classification $p_w(y|g, x) = \text{Bern}(y; h_w(gx))$, where the form of $h_w$ is known. Training the graphical model amounts to learning the unknown non-random parameters $w$. We train the graphical model using a dataset $S = (x_i, y_i)_{i=1}^n$. We denote $S_y = \{y_i\}_{i=1}^n, S_x = \{x_i\}_{i=1}^n$. We further assume that the $x_i$s are independent ($x_i \perp\!\!\!\perp x_j$), while the $y_i$s are conditionally independent ($y_i \perp\!\!\!\perp y_j|g$). Figure 8 depicts one such graphical model.

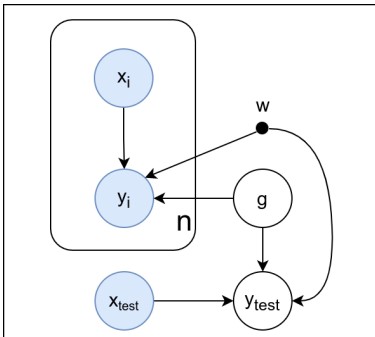

Figure 8: Probabilistic Graphical Model analogue. Circles represent random variables and arrows represent conditional probabilities. Filled circles are observed variables. Dots represent parameters.

We do not need to model the density of $x_i$s since during training we maximize the conditional likelihood (marginalizing over $g$), solving $\max_w \log p_w(S_y|S_x)$. Note that for $i \neq j$, we do not know that $y_i \perp\!\!\!\perp y_j|x_i, x_j$ when $g$ is not observed (i.e., we may have $y$s not D-separated in the graphical model). So, the above expression is not separable in the samples and thus not easily optimized. Explicitly writing the expectation over $g$ does not seem to make the optimization more convenient, since the expression becomes:

$$\max_w \log p_w(S_y|S_x) = \log \mathbb{E}_{g \sim p(g)} \prod_{i=1}^n p_w(y_i|x_i, g) \tag{25}$$

This is not ideally suited for stochastic optimization, because we cannot easily construct unbiased estimators of the gradient with respect to the parameters $w$. Additionally, the product of distributions may not be numerically stable as $n$ increases.

Since exact training is not convenient, we can train the graphical model using variational inference by maximizing a lower bound on this objective. First we need to approximate the posterior $p(g|S)$. To be consistent with the TRM analogy, we choose the variational posterior family to be the same as the transformation hypothesis space $\Omega$. In particular, since in TRM we sample independent transforms, this family contains distributions $Q \in \Omega$ for which $Q(g|y, x) = Q(g)$.

This might seem a restrictive family at first glance, however, since $g$ is a transformation acting on $x$, the transformed sample $gx$ is still dependent on the initial sample $x$. Only the *distribution* of the transformation $Q(g)$ is sample-independent. For example, this allows for a uniform distribution over rotations in some fixed range to be in the variational family, assigning the same probability to two different rotated vectors given the original vector, i.e., $p(R_\theta x|x) = p(R_\theta x'|x')$, where $R_\theta$ is the rotation matrix with angle $\theta$. On the other hand, a uniform distribution over rotations in some range that is dependent on the sample $x$ cannot be in the variational family.

Variational inference suggests finding the best approximate posterior in the variational family as measured by the Kullback-Leibler divergence between the variational posteriors $Q(g) \in \Omega$ and the true posterior $p(g|S)$. Finding both the optimal $w$ and the optimal $Q \in \Omega$ involves maximizing the following objective jointly over $w$ and $Q$:

$$\mathcal{L}(w, Q) = \log p_w(S_y|S_x) - KL(Q(g)||p(g|S))$$
$$= \log p_w(S_y|S_x) - \left( KL(Q(g)||p(g)) + \mathbb{E}_{g \sim Q(g)} \log \frac{p_w(S_y|S_x)}{p_w(S_y|S_x, g)} \right).$$

By cancelling terms, this further equals

$$\mathbb{E}_{g \sim Q(g)} \log p_w(S_y|S_x, g) - KL(Q(g)||p(g)) = \mathbb{E}_{g \sim Q(g)} \log \prod_{i=1}^{n} p_w(y_i|x_i, g) - KL(Q(g)||p(g))$$
$$= \sum_{i=1}^{n} \mathbb{E}_{g \sim Q(g)} \log p_w(y_i|x_i, g) - KL(Q(g)||p(g)). \tag{26}$$

We used the independence $g \perp\!\!\!\perp x_i$ for all $i \in [n]$ and $y_i \perp\!\!\!\perp y_j|g, x_i, x_j$ for $i \neq j$ from the graphical model. The objective is clearly a lower bound for equation equation 25 as the Kullback-Leibler divergence is non-negative and can be zero only if $p(g|S) \in \Omega$.

Using the prior on $g$ defined above as a uniform density on $G_\Omega$ (which has larger support than the variational posterior, thus the Kullback-Leibler term is well-defined), minimizing Equation equation 26 over $w, Q$ is equivalent to solving:

$$\max_{w, Q \in \Omega} \sum_{i=1}^{n} \mathbb{E}_{g \sim Q(g)} \log p_w(y_i|g, x_i) + H(Q). \tag{27}$$

where $H(Q) := -\mathbb{E}_{g \sim Q}[\log Q(g)]$ is the differential entropy of $Q$.

The equation above is similar to the one from Theorem 3.1.

1. The first term is the ETRM objective with loss $-\log p_w(y|g, x)$. For example, if the observation model is $y|g, x \sim \text{Bern}(h_w(gx))$, then $-\log p_w(y_i|g, x_i) = -[y_i \log h_w(gx_i) + (1 - y_i) \log(1 - h_w(gx_i))]$ which is the binary cross entropy loss, commonly used for binary classification.

2. The second term is a regularizer that promotes the selection of distributions with high entropy.

At test time, the Bayesian approach is to find $p_w(y_{test}|x_{test}, S)$, which results in averaging over the posterior of the transformations as follows:

$$p_w(y_{test}|x_{test}, S) = \mathbb{E}_{g \sim p(g|S)}[p_w(y_{test}|g, x_{test})], \tag{28}$$

where we used that $g \perp\!\!\!\perp x_{test}|S$ and $y_{test} \perp\!\!\!\perp S|g, x_{test}$ from the graphical model. Hence, if we use the variational posterior $Q^*$ found by optimizing equation 27 to approximate $p(g|S)$, and the model $h_{w^*}$ found by optimizing the same objective, we can construct the estimate:

$$\hat{p}(y_{test}|x_{test}, S) = \mathbb{E}_{g \sim Q^*}[p(y_{test}|h_{w^*}(gx_{test}))]$$

which is a direct analogue of using test time augmentation.

If the global variable $g$ were substituted by local variables $g_i$ for each $x_i, y_i$, then amortized variational inference in the resulting new graphical model would also result in an objective similar to the one from Theorem 3.1. However, the prediction step would be slightly different.

