# OpenReview forum: "Learning Augmentation Distributions using Transformed Risk Minimization"
_TMLR — Accepted by TMLR_

### Review · Reviewer_idNZ · 2023-03-23

**Summary Of Contributions:**

The paper proposes to learn a distribution over input transformations which will hopefully improve generalization. They make this tractable by making an approximation to the true "ideal" objective by pulling the expectation wrt the augmentations outside, and then parameterize a family of distributions of augmentations to optimize over using SGD.

**Audience:**

No

**Claims And Evidence:**

No

**Requested Changes:**

Why is this idea different from existing methods which learn augmentations, like AutoAugment?

Do you have any experiments on non-toy settings where this method does better than (or even matches) existing methods? This last point would not be necessary in my opinion if the work were doing something fundamentally new, but I don't believe it is.

**Strengths And Weaknesses:**

Unfortunately I don't think this idea contributes much to the task of generalization. First of all, the parameterized family of augmentations would have to capture the actual desired invariances in order for this to be helpful (e.g., using rotations on rotated MNIST), but this is a chicken-and-egg problem. We don't know what the correct family of distributions are---if we did, we could just apply it directly. Showing that this does something meaningful on toy data doesn't surprise me and I don't expect it to generalize.

I understand the idea of "learning" an augmentation distribution automatically, but this isn't a particularly inspired or creative idea, and it really doesn't seem too different from just trying out several augmentations and choosing the best one via a holdout set. Furthermore, it could even be _more_ amenable to overfitting---there's no real reason to expect this approach to improve generalization, and in fact I think this approach can actually _increase_ model complexity. I had this thought before seeing the experiments section, which confirm that existing augmentation methods do better quite consistently.

Finally, isn't this idea basically what AutoAugment already does? It learns a policy for augmentations based on the data and training process of the model, so it's already doing the same thing? I just don't see what this work does that is new or interesting, even as a proof-of-concept, not to mention the fact that empirically it doesn't seem to actually work.

---

> ### Author Response · Authors · 2023-05-25
> **Response to Reviewer idNZ (1/2)**
>
> We thank the reviewer for taking the time to review our paper. We would like to reply to the reviewer's comments and concerns.
>
> >  I understand the idea of "learning" an augmentation distribution automatically, but this isn't a particularly inspired or creative idea, and it really doesn't seem too different from just trying out several augmentations and choosing the best one via a holdout set.
>
> We are sorry that our idea does not come across as exciting; but we would like to assure the reviewer that for us it was both intriguing, exciting, and challenging to work on this project. It started out motivated by questions from practitioners in cancer imaging that asked us how to design practical data augmentation strategies. Since we did not have good answers, this is what motivated us to do this work.
>
> Trying out several augmentations and choosing the best one via a holdout set is not practically feasible when the augmentation space is expressive enough. The reason is that one first needs to discretize the continuous augmentations such as rotations, scalings, shearings, etc., and then create all different combinations of these finite sets. For example, if we discretize the parameter of a single augmentation family using $V$ bins and we combine $N$ such families (even in a fixed order) we would create $V^N$ candidate augmentations which also implies that we would need to train the model $V^N$ times. One would also need a large enough holdout set to perform this search which would harm training.
>
> This is the reason the field of automatic discovery of data augmentations was created,  first by the seminal paper of AutoAugment and later improved by many subsequent papers. Our method proposes a novel, differentiable parametrization of the augmentation space that is end-to-end trainable (together with the model) and avoids an exhaustive search of the augmentation space.
>
> > Finally, isn't this idea basically what AutoAugment already does? It learns a policy for augmentations based on the data and training process of the model, so it's already doing the same thing?
>
> Our approach is vastly different from AutoAugment. This is clearly illustrated by the fact that AutoAugment takes more than 5000 GPU hours to perform the policy search in CIFAR10 while we accomplish it in only 2.5 hours. To achieve this speedup we had to make several important contributions:
> 1. First, in contrast to AutoAugment we do not discretize the augmentation space. We propose a novel, expressive parametrization that is differentiable and thus end-to-end trainable with the model.
> 2. For this reason, we perform the search in a single training loop while AutoAugment requires multiple full model trainings which makes it impractical.
> 3. Moreover, to avoid overfitting we propose a novel regularization technique. By addressing the problem theoretically and providing a novel statistical framework we can reduce it to other well-studied statistical frameworks such as PAC-Bayes that provide us with a principled way to design the regularization term.
>
> > Furthermore, it could even be more amenable to overfitting---there's no real reason to expect this approach to improve generalization, and in fact I think this approach can actually increase model complexity. I had this thought before seeing the experiments section, which confirm that existing augmentation methods do better quite consistently.
>
> In the paper, we introduce a new regularization technique utilizing the well-studied PAC-Bayes theory that takes care of possible overfitting as is clearly evident in our experiments. In particular, from Table 1 one can observe that our method systematically surpasses the baselines as well as many state-of-the-art methods which include Augerino, DADA, and Faster AutoAugment. Only in the cases of Fast AutoAugment and Trivial Augment our method sometimes compares favorably and some other times not. However, in Fast AutoAugment the policy search is also three times slower than our method which introduces a very large overhead to the training time.

---

> > ### Author Response · Authors · 2023-05-25
> > **Response to Reviewer idNZ (2/2)**
> >
> >
> > > Do you have any experiments on non-toy settings where this method does better than (or even matches) existing methods? This last point would not be necessary in my opinion if the work were doing something fundamentally new, but I don't believe it is.
> >
> > 1. We added in Appendix A.6 an experiment on the SVHN dataset. In this experiment, we illustrate that precomputed policies from slow automatic augmentation methods like Fast AutoAugment do not transfer across even similar datasets (SVHN-train, SVHN-extra). Thus, a new search over the augmentation space is required for every new dataset or model which strengthens the argument in favor of faster automatic augmentation methods with good performance. To this end, we also compare against Augerino and DADA which we outperform.
> > 2. We also included an experiment on automatic augmentation discovery for the problem of Language Navigation in Section 5.5. In this problem, given a semantic map of the environment and an instruction provided in natural language, the goal is to output waypoints on the map that follow the instructions. The semantic maps are from real scans of Matterport3d. With this experiment, we show that our method can be easily incorporated into large-scale problems and benefit them in terms of performance.
> >
> > > First of all, the parameterized family of augmentations would have to capture the actual desired invariances in order for this to be helpful (e.g., using rotations on rotated MNIST), but this is a chicken-and-egg problem. We don't know what the correct family of distributions are---if we did, we could just apply it directly. Showing that this does something meaningful on toy data doesn't surprise me and I don't expect it to generalize.
> >
> > We are sorry that there might be some miscommunication here. In practice it is often the case that one believes that some form of geometric transformations (such as rotations or scalings) could capture invariances in the data. For instance in certain biological imaging applications, the tissues that are imaged could be in arbitrary rotations, and also the objects in the images could be scaled arbitrarily. However to actually use data augmentation, one needs to choose the specific parameters of rotations and scaling. If this is a new data set, such choices are not obvious. Our method is designed to automatically choose the parameters in such cases.
> >
> > We do not believe that there is a chicken-and-egg problem. Although discovering useful augmentations is hard, it is a well-defined problem. The reason is that one can incorporate in the search space any candidate transformation before one knows if it is an invariance. The goal of the method is to identify which of those transformations are indeed invariances (or more generally useful augmentations for the task).
> >  The need to pre-specify a hypothesis space or family of models is in alignment with standard machine learning.

---

### Review · Reviewer_tzE7 · 2023-04-04

**Summary Of Contributions:**

The authors tackle learning of data augmentation distributions.

They cast this as a risk minimisation problem, where the data augmentation is an additional model parameter; the goal is therefore to learn a data augmentation that generalizes well, i.e. that minimises the so-called *transformed population risk*. To this end, they minimize a PAC-Bayes-inspired regularization of the empirical risk of the augmented classifier. They also provide some simple yet insightful theoretical results on transformed risk minimization.

To optimise efficiently the augmentation distribution, they rely on a clever decomposition combined with the reparametrisation trick.

They show that, in practice, their approach can learn efficiently useful augmentations.



**Audience:**

Yes

**Broader Impact Concerns:**

I do not have particular concerns there.

**Claims And Evidence:**

Yes

**Requested Changes:**


**Thinks that could improve the paper:**

- Many previous approaches can be recast as TRM (in the sense that they try to minimise the transformed population risk), with sole specific choices of loss functions, and regularizer, for instance Augerino, Loraine et al. (2020), etc. It is perhaps a bit of a stretch to call TRM "a new theoretical framework" as the authors do on page 2 because of this. It would be interesting to list and comment such examples.  This would illustrate nicely the usefulness of the framework, and make the theory of Section 4 more broadly applicable.

- Doing more ablation-like experiments (see the "Strengths and Weaknesses" section). I think it would me more interesting to known which contribution are most useful in general, and not necessarily that the results are better than others.

- I feel like putting the related work section after the experiments harms the narration, I think putting it after the introduction would be much clearer

**Small clarifications:**

- I find it a bit confusing that the authors say "we aim to optimize [the empirical transformed risk]" just before Equation (3), whereas, as they mention later, actually optimizing it would lead to degenerate augmentations. I feel like I would be clearer to say that the ultimate goal is to optimize the transformed population risk, and that to that end, they optimise a regularized version of the empirical transformed risk.

- For MNIST, you mention that the learned range of the rotations was [-0.31,0.31], what about rotMNIST? It would be nice to do curves similar to those of Fig. 3 (left and center) but for the range of the rotations and not just for $\pi_i$-s, perhaps in the appendix.





**Strengths And Weaknesses:**

**Strengths**

- I think that the general way of posing the problem as Transformed Risk Minimisation, akin to vicinal risk minimisation, is elegant and clean. As I detail in the "Requested changes" section, I think it should be more clearly mentioned that this is essentially identical to previous approaches to learning data augmentations, but I think that none of these approaches wrote things in such a clean and simple way.

- The parametrization of augmentations, combined with unbiased gradient estimates, is quite clever, and could be useful.

- A big challenge when learning data augmentations by risk minimisation is regularization, and the PAC-based approach of the authors is a fresh perspective on these issues

**Weaknesses**

- The authors did miss a few recent works that tackle the same problem in similar ways. The idea of dealing with discrete augmentations with mixtures was previously used by Hataya et al. (2020) and in a more applied setting  by Raghu et al. (2022). These works relied on continuous relaxations of discrete random variables, whereas the authors here use a nice trick to derive unbiased estimates (Appendix A.3), so there is some novelty, but it is hard to assess it clearly.

- The PAC-based regularization is interesting, but not really well-motivated. Regularisation is clearly needed, as the authors argue, but why go in that direction? Doing a fair comparison (in the form of an ablation study, with grids of hyper-parameters) between this new regularization and simpler ones (e.g. the L2 regularization used by Benton et al.) would enable the community to assess the usefulness of this new regularization. There are also more complex approaches to avoir learning singular augmentation by overfitting, e.g. the line of work initiated by Lorraine et al. (2020), and followed by Hataya et al. (2020) and Raghu et al (2022) ; or in a Bayesian context, Schwöbel et al. (2022), Immer et al. (2022)

- It feels difficult to gain some clear scientific insights from the experiments. What are the new bricks designed by the authors that are most useful, is it the regularization, the way to compute gradients, the parametrization of augmentations?


**Additional references**


Immer et al., Invariance Learning in Deep Neural Networks with Differentiable Laplace Approximations, NeurIPS 2022

Lorraine et al., Optimizing Millions of Hyperparameters by Implicit Differentiation, AISTATS 2020

Hataya et al., Meta Approach to Data Augmentation Optimization, arXiv:2006.07965, 2020

Schwöbel et al., Last Layer Marginal Likelihood for Invariance Learning, AISTATS 2022

Raghu et al., Data Augmentation for Electrocardiograms, CHIL 2022

---

> ### Author Response · Authors · 2023-05-25
> **Response to Reviewer tzE7 (1/2)**
>
> We would like to thank the reviewer for providing a thorough review of our paper and for appreciating our novel formulation of the problem which allows us to use statistical tools like PAC-Bayes to derive the regularizer and permits the computation of unbiased gradient estimators.
>
> ##### **Regarding the novelty of the gradient estimation**:
>  We thank the reviewer for the additional references. We included them in the related work where we also discuss some differences with our approach. Our unbiased estimates are different from those of Hataya et al. and Raghu et al. Their approaches rely on a continuous relaxation of the discrete random variables that control the selection of augmentations and use a reparametrization trick as in [1],[2]. We avoid gradient estimators of this form as well as those relying on the RELAX estimator (DADA [3]) by directly calculating the derivatives and estimating them by sampling. In contrast to previous methods, our gradient estimators are unbiased and do not need variance reduction techniques.
> 1. Chris J. Maddison, Andriy Mnih, and Yee Whye Teh. "The concrete distribution: A continuous relaxation of
> discrete random variables". In International Conference on Learning Representations, 2017.
> 2. Eric Jang, Shixiang Gu, and Ben Poole. "Categorical reparameterization with gumbel-softmax". In 5th International Conference on Learning Representations, ICLR 2017.
> 3. Li, Yonggang, et al. "Differentiable Automatic Data Augmentation". Computer Vision -- ECCV 2020, edited by Andrea Vedaldi et al., Springer International Publishing, 2020, pp. 580–595.
>
> ##### **Regarding the motivation behind PAC-Bayes**:
> We do not necessarily think that this is the only choice for regularization but we do think it is well-motivated due to the following reasons:
> 1. PAC Bayes theory is known to lead to some of the sharpest generalization bounds for large-scale neural network models,
> 2. In our framework the augmentations are chosen at random, which mirrors the practical use of augmentations. PAC Bayes theory is well known to naturally handle randomized predictors.
> These are the two factors that were our main motivation for going in this direction.
> Moreover, we include in Appendix A.7 a Bayesian Perspective of the upper bound that we derived using PAC-Bayes theory (which is a frequentist perspective). We show that maximizing a variational lower bound of a graphical model analog leads to a similar objective as ours. This could potentially unify our framework with other Bayesian approaches mentioned by the reviewer that we discuss in the related work.
>
> ##### **Regarding ablations on the regularization and parametrization**:
> We added ablations in Table 2 where we evaluate the method using:
>  - First the L2 regularizer from Benton [1] with the SCALE parametrization and
>  - Second the parametrization from Benton [1] with our PAC-Bayes-based regularize. \
> We observe that both our PAC-Bayes regularizer and our expressive parametrization contribute to the performance.  Our proposed training objective and parametrization allow us to avoid gradient estimators that are known to suffer from high bias or high variance. Instead, we compute the gradients directly and estimate the expectation from Monte-Carlo sampling thus getting an unbiased estimator with well-behaved variance. Different parametrizations or objectives like in Benton, DADA, and Hataya et.al would not permit such unbiased estimators.
> 1.  Gregory Benton, Marc Finzi, Pavel Izmailov, and Andrew G Wilson. Learning invariances in neural networks from training data. In Advances in Neural Information Processing Systems, 2020.

---

> > ### Author Response · Authors · 2023-05-25
> > **Response to Reviewer tzE7 (2/2)**
> >
> >
> > ##### **Regarding the connection of TRM with other methods**:
> > We do think that our theoretical framework is novel. We agree that special cases have been discussed before, but we think that it is valuable to consider it at this level of generality because it helps with understanding and connecting disparate research contributions
> > Some of the approaches mentioned by the reviewer use similar techniques but not all their choices can be expressed inside our TRM framework. We add the references in the related work and discuss differences. As we mentioned above, we also added a Bayesian perspective of our training objective that was derived as an upper bound of the population risk, which we recast as a variational lower bound to a maximum likelihood objective. This viewpoint might connect some of the Bayesian approaches in the area as well.
> >
> > ##### **Regarding the Section on Related Work**:
> > We chose to put the related work after introducing the framework since this allows us to delve deeper into the similarities and differences between our method and previous approaches. If we change the order then we would not be able to do a detailed discussion on particular choices that have not been introduced yet in the paper. We extended the discussion in the related work incorporating your suggestions, but if you still insist that changing the order would enhance clarity we will make this change.
> >
> > ##### **Regarding small clarifications**:
> > 1. To improve clarity we changed the phrasing in this section. We added: "Given a finite dataset, empirical transformed risk minimization (ETRM) would aim to optimize ..." and then explained "However, as we discuss in Section 3.1 this training objective is not a good estimate of the population risk Eq.2 as it collapses to trivial distributions. In the next section, we discuss how to optimize the population risk by proposing an upper bound to Eq.2 that is computable from the training data and takes care of this problem".
> > 2. Regarding the diagrams with the range of parameters we added two diagrams corresponding to CIFAR10/100 in the Appendix.
> >
> > ##### **Regarding additional changes in the paper**:
> > 1. Following the suggestions of other reviewers we also added in Appendix A.6 an experiment on SVHN dataset. In this experiment we illustrate that precomputed policies from slow automatic augmentation methods like Fast AutoAugment do not transfer across even similar datasets (SVHN-train, SVHN-extra). Thus, new search over the augmentation space is required for every new dataset or model which strengthens the argument in favor of faster automatic augmentation methods with good performance. To this end, we also compare against Augerino and DADA which we outperform.
> > 2. We also included an experiment on automatic augmentation discovery for the problem of Language Navigation in Section 5.5. With this experiment, we show that our method can be easily incorporated into large-scale problems and benefit them in terms of performance.
> > 3. Lastly, we added comparisons against other methods like DADA in CIFAR10/100 in Table 1.

---

### Review · Reviewer_QAqa · 2023-05-03

**Summary Of Contributions:**

​
This paper first proposes a new risk minimization (RM) objective, i.e., transformed risk minimization (TRM), that accounts for both the data distribution and the transformation distribution. The authors then derive an upper bound for the population risk in terms of the empirical risk and regularization terms. They also provide a practical algorithm, stochastic compositional augmentation learning (SCALE), for joint optimization over the model and data augmentation parameter space. They show theoretically that TRM can recover the distributional invariance and the corresponding optimal model under certain conditions. Empirically, SCALE is comparable to state-of-that-art methods and recovers the rotational invariance on rotMNIST dataset.

**Audience:**

Yes

**Claims And Evidence:**

Yes

**Requested Changes:**

1. There are a few works that use gradient-based methods and jointly learn the model and data augmentation in one training loop [1] [2]. How is SCALE compared with them in terms of classification performance and training time cost (GPU hours)?

2. The paper does not include experiments on larger datasets such as SVHN and ImageNet. It is good to see whether SCALE still has good generalization when the dataset becomes larger.
​
[1] Lin, Chen, et al. "Online hyper-parameter learning for auto-augmentation strategy." Proceedings of the IEEE/CVF International Conference on Computer Vision. 2019.
​
[2] Li, Yonggang, et al. ‘Differentiable Automatic Data Augmentation’. *Computer Vision -- ECCV 2020*, edited by Andrea Vedaldi et al., Springer International Publishing, 2020, pp. 580–595.

**Strengths And Weaknesses:**

--- The strengths of this paper are:
​
1. The paper is clearly written and easy to follow.

2. The proposed TRM framework extends the standard RM framework to factor in data transformations. It provides a practical solution to the long-standing problem of choosing proper data augmentation strategies with incomplete knowledge only.

3. The SCALE algorithm is simple to implement and incorporate into standard training algorithms. Experiment results show that it is comparable in terms of classification performance to those methods that are more computationally expensive. In addition, the models trained by SCALE also tend to be well-calibrated.
​
--- The weaknesses (mostly nitpicking) are:
​
1. The regularizer involves some hyperparameter (e.g., $\lambda_{\text{reg}}$ that requires extra tuning for each dataset, which can be time-consuming.

2. Lack of explanation of the necessity of linearly decay of regularizer $\lambda_{\text{reg}}$ Does it imply that the joint training does not converge when using a fixed $\lambda_{\text{reg}}$?

---

> ### Author Response · Authors · 2023-05-25
> **Response to Reviewer QAqa**
>
> We would like to thank the reviewer for taking the time to provide a constructive review of our paper and for recognizing our contribution to the problem of automatic data augmentation discovery.
>
> #### **Regarding comparisons against DADA [2], OHL-Aug [1]**:
>   - **DADA**: Following the reviewer's suggestion, we included a comparison with DADA [2] in Table 1. For a fair comparison, we evaluate DADA on the same set of base augmentations (geometric transformations) as our method and search for the best policy over the whole CIFAR 10/100.  We observe that SCALE outperforms DADA in both CIFAR10/100 and is also faster.
>   - **OHL-Aug**: The goal of our method is to propose a fast alternative to AutoAugment that performs the search over the augmentations with only a small overhead over the standard training time. Our comparisons are against methods whose overhead is in a similar order of magnitude. However, OHL-Auto-Aug takes nearly 83 GPU hours to find good augmentations in CIFAR10 which is over 30 times slower than our method. We also mention these differences in the related work.
>
> ##### **Regarding experiments on larger datasets**:
>  -  **SVHN**: We added the experiment on SVHN in Appendix A6. In this experiment, we illustrate that the policies computed by slower methods like FAST-AA are not transferrable even on similar datasets (SVHN-train and SVHN-extra). Thus a new search over the augmentations is needed for every new dataset which strengthens the argument for faster methods that perform well. We also compare against such methods (DADA, Augerino) and show that we outperform both.
> -   **Language Navigation**: Following the suggestion of reviewer idNZ we show that one can incorporate SCALE easily into large-scale problems and benefit them in terms of performance. We applied SCALE to a Language Navigation problem where given a semantic map and an instruction the goal is to predict waypoints in the map that follow the instruction. We show that the problem can be benefited from the addition of SCALE and do an ablation study on the range of $\lambda_{reg}$ at the same time.
>
> ##### **Regarding the linear decay of $\lambda_{reg}$**:
> The linear decay of $\lambda_{reg}$ is equivalent to a linear decay in the learning rate. We apply the same scheduling to the first term of the loss. This is a standard technique for making training more robust to hyperparameters. It is also used in the baseline methods we compare with.
>
> #### **Regarding tuning $\lambda_{reg}$**:
> Using the linear scheduling that we discussed makes the method relatively robust to $\lambda_{reg}$. In the Appendix A.2.3 in Optimization over the hyperparameters $\sigma,\lambda$ we also show that by optimizing the PAC-Bayes bound w.r.t. $\lambda_{reg}$ we get the order of the hyperparameter, $\Theta(1/\sqrt{n})$. We use this value as an estimate for the initialization. As the number of samples increases this estimate gives a tighter upper bound. This can also be seen in Figure 6, where the empirical best value for $\lambda_{reg}$ is close to the theoretical one.

---

### Decision · Action_Editors · 2023-07-03

**Recommendation:** Accept with minor revision

**Comment:**

Please include the following changes in the final revision:
1. Make it clear that you are not claiming that the gradient estimators you describe are novel and cite the relevant prior work. The reparameterization of the uniform distribution for continuous augmentation parameters has already been used in Augerino, and the mixture-based gradient estimator for the discrete case is an application of Local Expectation Gradients [1].
2. Explain the exact setup and motivation for annealing the regularization term in the loss. Currently the paper seems to suggest that you are linearly reducing the weight of the regularizer in the loss to zero during training, but in a response to a reviewer, you said that the linear decay also applied to the other term in the loss function and thus is equivalent to using linear learning rate decay. If this is indeed the case, why is this necessary if you are already using (cosine) learning rate decay as described in the paper?

[1] Local Expectation Gradients for Black Box Variational Inference, Michalis K. Titsias, Miguel Lázaro-Gredilla, NeurIPS 2015

**Audience:**

While there was no agreement between reviewers as to whether there is an audience for this paper, I expect this paper to be of interest to at least some researchers working on learning augmentations.

**Claims And Evidence:**

The paper extends the risk minimization framework to allow learning distributions over augmentations jointly with model parameters from (training) data. To rule out trivial solutions for augmentations, the authors derive a regularizer based on PAC-Bayes theory. The performance of the method is mostly competitive with the state-of-the-art, especially when taking into account training times.
The paper is fairly well written, and provides a clean formalization of the problem and a convenient parameterization of augmentation distributions.

---

> ### Author Response · Authors · 2023-07-25
> **Camera Ready Revision**
>
> We would like to thank all the reviewers and the action editor for providing valuable feedback on improving our paper. In the camera ready version we incorporated the suggested minor revisions:
>  - In Section 3.3 we added the necessary references, suggested by the action editor, on the computation of the gradients with respect to the augmentation parameters
>  - We revised Section A.4 in the Appendix to describe extensively the training details. Specifically, we explain how we use linear learning rate decay for the optimization of the augmentation parameters, and cosine annealing of the learning rate for the optimization of the network parameters.
>
> We also included the code in the supplementary material.